# Identifying hotspots of greenhouse gas emissions from drained peatlands in the European Union

Quint van Giersbergen [1] ✉, Alexandra Barthelmes[2], John Couwenberg[2], Kristiina Lång[3], Nina Martin[2], Cosima Tegetmeyer[2], Christian Fritz[1,4,5] & Franziska Tanneberger [2,5] ✉

Greenhouse gas (GHG) emissions from drained peatlands account for about 7% of the total anthropogenic GHG emissions in the European Union (EU). Yet, a lack of high-resolution spatial data hampers targeted mitigation. We combined soil and land use data to generate detailed maps of land use, GHG emissions, and emission hotspots for EU+ peatlands. Undrained peatlands and those drained for forestry dominate at high latitudes, while drained Grassland and Cropland prevail around latitudes 50°–55°. Four main emission hotspots emerge: the North Sea region, eastern Germany, the Baltics together with eastern Poland, and north Ireland. The North Sea region is the largest, accounts for 20% of EU+ peatland emissions on just 4% of the peatland area. Our findings highlight the urgency of reducing emissions from drained peatlands to meet EU climate targets and reveal substantial underreporting in National UNFCCC inventories, amounting to 59–113 Mt $CO_2$e annually. Our analysis provides a robust and spatially explicit evidence base for policymakers to prioritize peatland rewetting to reduce GHG emissions.

In its Sixth Assessment Report, the Intergovernmental Panel on Climate Change states that limiting warming to around 1.5 °C requires anthropogenic greenhouse gas (GHG) emissions to peak before 2025[1] and that steps towards system-wide transformations to secure a net-zero, climate-resilient future are immediately taken[2]. One important source of anthropogenic GHG emissions is found in drained peatlands[3]. In natural, undrained peatlands, decomposition of dead plant material is incomplete, and peat (and carbon) accumulate. The World's peatlands store an amount equivalent to two-thirds of the total (current) atmospheric carbon (C)[4]. When peatlands are drained, the peat oxidizes, resulting in the release of carbon dioxide ($CO_2$). Global emissions from peatland degradation are estimated at 2 Gt $CO_2$e per year (without fires), which amounts to 5% of all anthropogenic emissions[3]. If GHG emissions from drained peatlands globally continue at the current rate, this could consume 12–41% of the GHG emission budget for keeping global warming below +1.5 to +2 °C[5].

In the European Union (EU), peatlands make up for almost 6% of the total land area and are present in every country except Malta[6]. Most of them are concentrated in the northern (boreal and temperate) lowlands. Half of the peatlands in the EU are degraded due to various human activities, most often linked to artificial drainage[3,7]. Most of the peatland drainage was done for agriculture, forestry, and peat extraction. Drained peatlands currently emit up to 230 Mt $CO_2$e per year in the EU and 580 Mt $CO_2$e per year in entire Europe[3,8] and are estimated to contribute some 5% to the total EU anthropogenic GHG emissions[9].

Accurate GHG emissions reporting is essential for the development of policy measures to reduce these emissions[10]. It enables

[1]Department of Ecology, Radboud Institute for Biological and Environmental Sciences (RIBES), Radboud University, Nijmegen, The Netherlands. [2]Institute of Botany and Landscape Ecology, Greifswald University, Partner in the Greifswald Mire Centre, Greifswald, Germany. [3]Natural Resources Institute Finland (Luke), Jokioinen, Finland. [4]Integrated Research on Energy, Environment and Society (IREES), University of Groningen, Groningen, The Netherlands. [5]These authors jointly supervised this work: Christian Fritz, Franziska Tanneberger. ✉e-mail: Quint.vangiersbergen@ru.nl; tanne@uni-greifswald.de

policies and projects to focus appropriately on areas with high GHG emissions, including drained peatlands. Therefore, all parties included in Annex 1 to the United Nations' Framework Convention on Climate Change (UNFCCC), including the EU and its individual member states, are obliged to report their GHG emissions in annual inventories. National Inventory Submissions (NIS) must be based on IPCC emissions reporting guidelines (including the 2013 Wetlands Supplement for managed organic soils/peatlands[11]) and contain two parts, namely the Common Reporting Tables and the annual National Inventory Document (used to be named Common Reporting Format, CRF and National Inventory Report, NIR). In the EU, emissions from drained peatlands contribute considerably to both the effort sharing sector[12] and the LULUCF sector[13,14] of total emissions regulated under the EU climate and energy framework.

There are two main shortcomings in reporting GHG emissions from peatlands. Firstly, the reporting of the GHG emissions the EU level is done via a bottom-up approach. Each country reports by itself, which makes the process prone to underestimates[10], e.g., by under-reporting the drained peatland area ("activity data"), using outdated emission factors (EFs; from earlier IPCC guidelines), not including all relevant GHGs emitted from drained peatlands ($CO_2$, $CH_4$, $N_2O$), or not reporting emissions from ditches. A comparison between the peatland area reported by the EU member states in 2017 and the data of the Global Peatland Database (GPD, https://greifswaldmoor.de/global-peatland-database-en.html) revealed that many EU member states under-reported their drained peatland area at that time[15]. As a result, only 92 Mt $CO_2$e per year were reported to the UNFCCC for agriculturally used peatlands in the EU in 2021 (Cropland and Grassland), compared to 167 Mt from an improved assessment based on more comprehensive and accurate area data and updated IPCC EFs[16]. Secondly, the reporting under the UNFCCC does not require detailed spatial information showing where the peatlands are located and where exactly in the EU peatland emissions are high. Yet, this information is crucial for policy-makers to develop climate-smart land use policies for GHG mitigation[17,18], as well as to implement cost-efficient rewetting and restoration measures.

Already more than 20 years ago, McClain et al. (2003) suggested that easy wins for the climate could be achieved by first focusing on hotspot areas[19]. Unfortunately, GHG emissions from peatlands were hardly reduced over the past years[20], despite the increasing accuracy in reporting[16] and rising awareness about the potential for climate mitigation by rewetting drained peatlands. Apparently, both policymakers and society still need to be made more aware of GHG emissions from drained peatlands. A promising step forward is to inform about peatland emissions across the EU based on improved spatial maps. The aims of this work are (1) to increase the accuracy in EU GHG reporting from peatlands by providing a detailed peatland GHG emissions map, and (2) to promote targeted policy measures for GHG mitigation by introducing an EU-wide peatland GHG hotspot map. Moreover, the resulting data can stimulate the exchange between EU and IPCC bodies and their member states on the quality of national reporting to the UNFCCC.

## Results

### Land use distribution

Grasslands on peatland dominate around the 50–55° N latitude while Forest Land is most prevalent at higher latitudes (Fig. 1). Around 75% of all Croplands (1302 kha) on peatland are located in Germany (391 kha), the United Kingdom (198 kha), Lithuania (158 kha), Poland (153 kha), Finland (138 kha), Hungary (135 kha) and the Netherlands (127 kha; Table 1 and Supplementary Table S.1). The largest areas of Grasslands on peatland can be found in Germany (954 kha), the United Kingdom (720 kha), Poland (558 kha), Ireland (432 kha), Iceland (283 kha) and the Netherlands (254 kha), together covering 75% of all Grasslands on peatland (Supplementary Table S.2). Most Forest Land on peatlands

(> 85% = 11680 kha) is located in Finland (5830 kha), Sweden (3897 kha), Poland (864 kha), Estonia (547 kha), Lithuania (280 kha) and Latvia (263 kha). The highest proportion (> 85%) of undisturbed peatland is found in Northern Europe countries, together with the United Kingdom, Ireland, and Iceland (Supplementary Table S.3).

This study estimates a larger total peatland extent for most countries compared to NIS 2023 using the updated peatland distribution map, which has been integrated into the new European Wetland Map (Tegetmeyer et al. 2025). Most countries seem to have a greater spatial extent of Cropland and Grassland on peatland than the areas reported to the UNFCCC (2023), only 6 out of 27 countries (and 8 out of 37 EU+ countries, see Table 1) report higher agricultural activity on peatland in their NIS than found in this study. Overall, the total Cropland and Grassland on peatland in the study area is 4% compared to the national reports. The estimates of total Forest Land agree well, although our area estimate of drained Forest Land is substantially higher. We arrive at a substantially higher total Forest Land area in Poland, but smaller Forest Land area in Norway, the United Kingdom, and Latvia (Table 1).

In this study, the area of drained peatlands used for agriculture and forestry often shows a deviation greater than 20% from corresponding values reported in the 2023 NIS of the respective EU countries. The area estimates for Cropland, Grassland, and Forest Land are largely aligning only for Germany, Finland, The Netherlands, and Iceland and some low peatland coverage countries such as the Czech Republic, Bulgaria, Slovenia, and Portugal (Table 1). In general, the data for Forest Land is more consistent than for Cropland or Grassland. Some 26 countries have similar coverage ( ± 20% in NIS 2023 and this study) of Forest Land, which accounts for 86% of the total EU+ Forest Land category on organic soils (Supplementary Table S.4). For Cropland and Grassland, the two estimates agree well in 18 countries, but these countries only account for 42% of the total EU+ Cropland and Grassland categories (Supplementary Table S.2).

### Peatland emissions

The highest GHG emissions from drained peatlands are observed between latitudes 50° and 55° N (Fig. 1B). Further north, larger areas of low-emitting peatlands are concentrated in Finland, Sweden, and Scotland, because here a substantial proportion of peatlands remains little affected by drainage, and emissions from boreal forested peatlands are low (Supplementary Table S.1). In Estonia, Latvia, Lithuania, Ireland, and the United Kingdom there are considerable areas of low-emitting peatlands, mostly shallow-drained Grasslands. Our summation of GHG emissions from peatlands yields 232 ± 56 Mt $CO_2$e for the EU, double the 119 Mt $CO_2$e reported by the EU countries to the UNFCCC (Table 1). GHG emissions from drained peatlands in the EU and EU+ contribute substantially (7.4 % and 7.5%, respectively) to the total EU and EU+ anthropogenic GHG emissions[20].

Our spatial GHG emission analysis suggests higher agricultural GHG emissions for the majority of countries (28 out of 37), than reported to the UNFCCC in 2023 (Table 1). Differences primarily originate from applying out of date EFs for Cropland and Grassland and less to underestimates in drained peatland area. Looking at EFs (emissions per hectare), 19 out of 37 countries use factors close to the IPCC default factors for agricultural land (Cropland + Grassland; indicated by [a] and [b] in Table 1). Peatland-rich countries with the largest emission deviations between NIS and our analysis are the UK, Hungary, Poland, Lithuania, The Netherlands, Ireland, France, and Estonia.

Based on this study, EU Forest Land emissions from drained peatland (Table 1; 103 Mt $CO_2$e) are substantially higher than those reported in NIS 2023 (25.2 Mt $CO_2$e). We see two main reasons (playing out for individual countries): underestimation of the peatland extent in the first place, as well as an inappropriate choice of EFs, or an incomplete coverage of gases ($CO_2$, $CH_4$, $N_2O$) and DOC export in NIS. Only Finland, Sweden, and Denmark have used EFs comparable to IPCC[11].

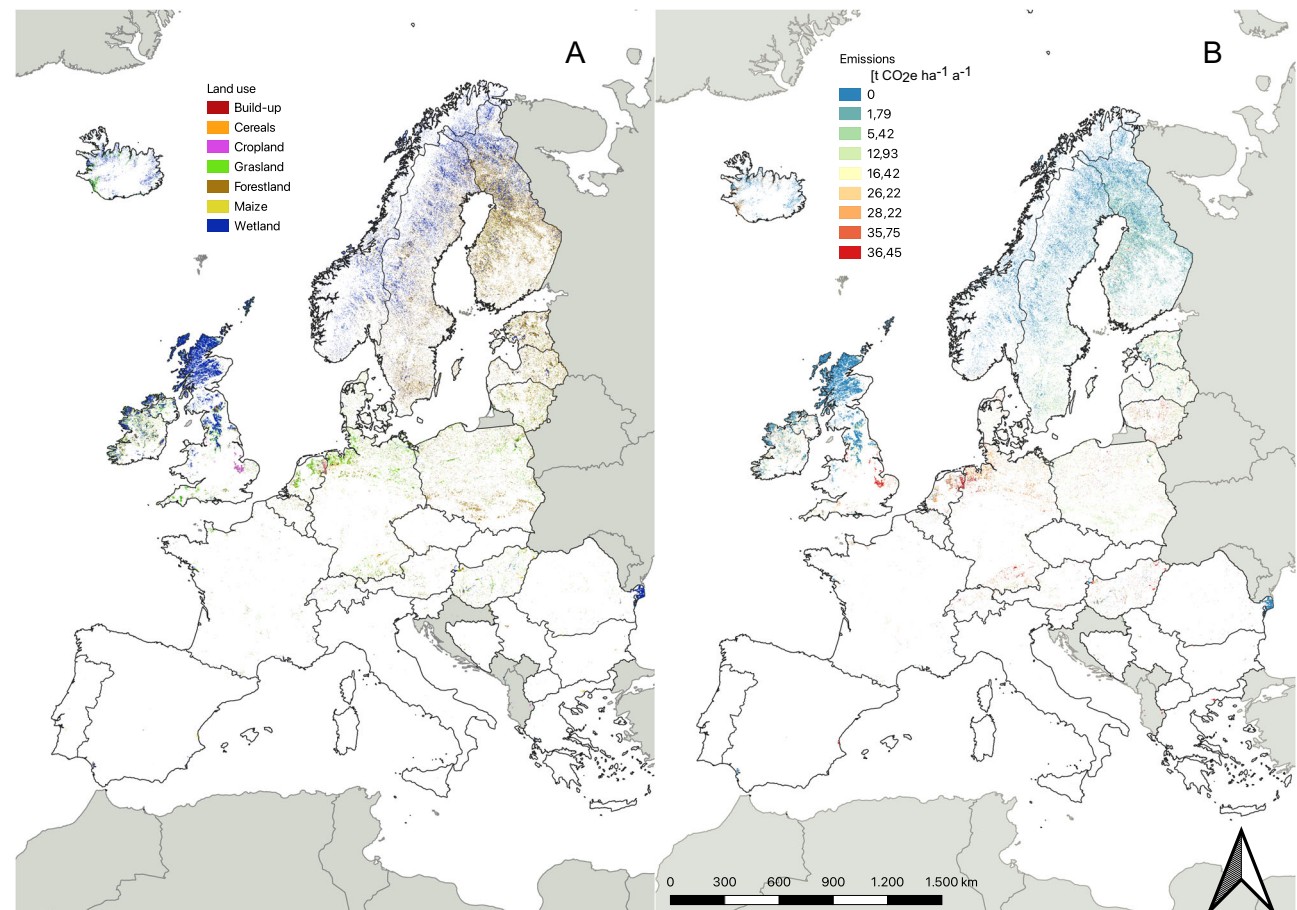

**Fig. 1 | Land use and greenhouse gas emissions accross EU+ peatlands.** Land use map (**A**) and greenhouse gas (GHG) emissions map (**B**, tons of $CO_2e$ $ha^{-1}a^{-1}$) of EU+ peatlands. To enhance visibility, certain land uses are aggregated as shown in Supplementary Table S1, and the resolution is downscaled to 1 km² using nearest neighbouring resampling, which shows that Grassland and Cropland are mainly located in the 50–55° latitude, whereas in the Baltic and Scandinavian countries Forest Land and Wetlands are a dominant type of land use.

The other countries have used lower EFs, on average 3 tonnes of $CO_2e$ $ha^{-1}$ $a^{-1}$, and some countries in the Balkan used very low EFs, referring to boreal EF estimates. The UK and Germany use rather low Forest land EFs, but these are actually rather well corroborated.

## Validation against regional mapping

The comparison of our results with regional peatland GHG emission maps from Brandenburg and Mecklenburg-Vorpommern shows a good fit in terms of mean emissions per region, with a correlation of around 0.64 and a MAE around 4. The mean difference for Branden-burg is only 2.8 t $CO_2e$ $ha^{-1}$ $a^{-1}$, and for Mecklenburg-Vorpommern 4.3 t $CO_2e$ $ha^{-1}$ $a^{-1}$ (Fig. 2 and Supplement Fig. S.2). The areas dominated by Grassland and Cropland (higher emitting) seem to agree well, whereas Forest Land dominated regions, (the lower emitting regions) exhibit higher emission in our estimate (Fig. 2 and Supplementary Fig. S.2).

## Peatland hotspot map

Our spatially explicit analysis revealed GHG emission hotspots across Europe. Peatland GHG emission hotspots (Fig. 3, indicated in purplish color = top-right four grids of legend matrix A) are responsible for 39% of the total peatland GHG emissions, despite covering only 17% of the total peatland area (top-right four grids of legend matrix B). The region with the highest relative emissions is situated in north-western Germany and north-eastern part of the Netherlands, contributing to 14.2% of the total EU+ peatland emissions. The western part of the Netherlands (1.1% of EU+ total), south-eastern England (1.9%) and north-eastern Germany (4.8%) also emerge as pronounced GHG hotspots,

while collectively covering just 6.3% of the total EU+ peatland area (Fig. 3). A second group of GHG emission hotspots is found in eastern Poland, northern England (UK), north-western Ireland, and in the Baltic states.

Regions with high cumulative GHG emissions from peatlands (cumulative, because there is a large area of peatlands) include the central European Plain, the Alpine foreland in Germany, as well as Hungary and Romania. Blue coloured hotspots indicate a high density of peatlands (peatland area hotspots in Fig. 3), which occur especially in Finland and Sweden, covering 4.2%, 3.6%, and 3.2% of the EU+ peatland area, but emitting only 1.4%, 2.7%, and 0.3% of the EU+ peatland GHG emissions, respectively.

## Discussion

This study provides spatially explicit information on peatland GHG emissions on the European scale. This information is relevant both for climate mitigation policy-making and policy implementation. Our comprehensive approach uses high-resolution land use data, a newly developed peatland map, and both country-specific analyses and cross-country regional assessments. Our analysis shows the benefits of integrating multiple spatial data products to estimate the density and distribution of peatland GHG emissions. Our results raise serious concerns about the overall under-reporting of peatland GHG emis-sions in the National GHG Inventory Submissions of EU+ countries to the UNFCCC ("NIS"), which amounts to 105–132 Mt $CO_2e$ annually (Table 1). This amount of under-reported emissions is roughly equivalent to the annual emissions from EU air traffic[21]. Main reasons

**Table 1 | Comparison of peatland area and emissions between this study and national inventories**

| | Agriculture | | | | | Forest Land | | | | | | | | Total | |
|---|---|---|---|---|---|---|---|---|---|---|---|---|---|---|---|
| | Area [kha] | | Emissions [kt] | | | Area [kha] | | | | Emissions [kt] | | | | Emissions [kt] | |
| | NIS | This Paper | NIS | This Paper | | NIS | | This Paper | | NIS | This Paper | | | NIS | This Paper |
| Country | Table 4B + 4C | Total | Table 3D + 4B + 4C | Pixel Area | StErr. | Table 4A | Table 4II | Forest Land | Pixel Area | Table 4A | Forest Land | Pixelsum | StErr. | Drained | Pixel sum |
| Germany | 1296 | 1346^a | 44165 | 37609^a | 7542 | 278 | 278 | 260^a | 558 | 808 | 3365 | 7213 | 1824 | 44973 | 44822† |
| The Netherlands | 341 | 381^a | 6220 | 11213 | 2296 | 20 | 4 | 18^a | 52 | 72 | 237 | 671 | 170 | 6293 | 11883 |
| Finland | 342 | 322^a | 9839 | 9790^a | 3421 | 5963 | 4313 | 5830^a | 5830 | 10653 | 18214 | 18214 | 6954 | 20492 | 28007 |
| Czech Republic | 0 | 5^a | 0 | 148 | 30 | 19 | NO | 15^b | 22 | 0 | 196 | 289 | 73 | 0 | 437 |
| Bulgaria | 3 | 5^a | 98 | 174^b | 36 | 0 | NO | 0^a | 2 | 0 | 6 | 20 | 5 | 98 | 194 |
| Slovenia | 4 | 4^a | 97 | 111^a | 23 | 1 | NO | 2^a | 3 | 0 | 27 | 43 | 11 | 97 | 154 |
| Portugal | 0 | 4^a | 0 | 139 | 29 | NO | NO | 1^a | 2 | 0 | 10 | 27 | 7 | 0 | 166 |
| Croatia | 3 | 3^a | 107 | 94^a | 20 | NO | NO | 10 | 13 | 0 | 124 | 171 | 43 | 107 | 265 |
| Luxembourg | 0 | 1^a | 0 | 23 | 5 | NO | NO | 0^a | 1 | 0 | 0 | 8 | 2 | 0 | 31 |
| Slovakia | 0 | 1^a | - | 13^a | 3 | NO | NO | 0^a | 1 | 0 | 3 | 14 | 4 | 0 | 28 |
| Andorra | - | 0^a | - | 0^a | 0 | - | - | 0^a | 0 | - | 0 | 1 | 0 | - | 1 |
| Cyprus | 0 | 0^a | 0 | 0^a | | NO | NO | 0^a | 0 | 0 | 0 | 0 | | 0 | 0 |
| Poland | 957 | 711 | 8633 | 18123 | 3486 | 340 | NA | 864 | 1408 | 849 | 1166 | 18208 | 4605 | 9482 | 36331 |
| Ireland | 339 | 445 | 8954 | 11630^b | 2347 | 457 | 414 | 442^a | 422 | 2998 | 5460 | 5460 | 1381 | 11952 | 17090 |
| Lithuania | 134 | 319 | 724 | 9806 | 2007 | 303 | 154 | 280^a | 485 | 411 | 3618 | 6265 | 1585 | 1135 | 16071 |
| Hungary | 0 | 307 | 0 | 9123 | 1850 | 6 | NO | 40 | 115 | 62 | 515 | 1487 | 376 | 62 | 10610 |
| Latvia | 166 | 123 | 2867 | 3597^b | 723 | 379 | 379 | 263 | 706 | 1207 | 3406 | 9130 | 2309 | 4074 | 12727 |
| Denmark | 172 | 101 | 5282 | 2971^b | 623 | 37 | 19 | 20^a | 80 | 204 | 260 | 1037 | 262 | 5486 | 4007 |
| Estonia | 83 | 110 | 796 | 3174 | 651 | 600 | 288 | 547^b | 780 | 1052 | 7071 | 10088 | 2551 | 1848 | 13262 |
| France | 13 | 84 | 21 | 2395 | 493 | NO | NO | 33 | 75 | -42 | 424 | 965 | 244 | -21 | 3360 |
| Romania | 8 | 75 | 229 | 2359^b | 491 | 3 | NO | 11 | 33 | 24 | 149 | 424 | 107 | 254 | 2783 |
| Austria | 13 | 72 | 376 | 2156^b | 442 | NO | NO | 40 | 67 | 0 | 519 | 867 | 219 | 376 | 3023 |
| Sweden | 165 | 53 | 4039 | 128^b | 267 | 4433 | 988 | 3897^a | 3897 | 6915 | 21997 | 21997 | 6272 | 10954 | 23278 |
| Belgium | 3 | 39 | 89 | 1149^b | 238 | NO | NO | 17^a | 49 | 0 | 217 | 638 | 161 | 89 | 1787 |
| Spain | 0 | 20 | 0 | 705 | 147 | NO | NO | 1^a | 5 | 0 | 17 | 68 | 17 | 0 | 774 |
| Greece | 7 | 15 | 280 | 540^b | 113 | - | - | 1^a | 2 | - | 14 | 27 | 7 | 280 | 566 |
| Italy | 24 | 1 | 972 | 29 | 6 | - | - | 0^a | 2 | 0 | 6 | 21 | 5 | 972 | 49 |
| Total EU | 4074 | 4545^a | 93787 | 128352 | 27288 | 12839 | 7188 | 12570^a | 14606 | 25215 | 77021 | 103351 | 29193 | 119002 | 231702 |
| Iceland | 347 | 304^a | 7898 | 8736^a | 1841 | 4 | 4 | 0^a | 7 | 9 | 90 | 90 | 23 | 7907 | 8825 |
| Norway | 70 | 81^a | 2463 | 2964^a | 618 | 708 | 133 | 263 | 263 | 1483 | 3403 | 3404 | 861 | 3946 | 6358 |
| Serbia | - | 6^a | - | 209 | 44 | - | - | 6^a | 7 | - | 81 | 96 | 24 | - | 305 |
| North Macedonia | - | 2^a | - | 22 | 5 | - | - | 0^a | 0 | - | 0 | 1 | 0 | - | 23 |
| Montenegro | - | 1^a | - | 52 | 11 | - | - | 1^a | 1 | - | 7 | 15 | 4 | - | 67 |
| Liechtenstein | 0 | 1^a | 0 | 0^a | 0 | - | NO | 0^a | 0 | 0 | 0 | 0 | 0 | 0 | 0 |
| United Kingdom | 1669 | 917 | 11981 | 22216 | 4136 | 455 | 451 | 305 | 305 | 461 | 3946 | 3946 | 998 | 12441 | 26162 |
| Switzerland | 17 | 51 | 682 | 1591 | 334 | 4 | 0 | 11 | 21 | 1 | 138 | 265 | 67 | 683 | 1856 |
| Albania | - | 17 | - | 605 | 127 | - | - | 1^a | 1 | - | 9 | 15 | 4 | - | 620 |
| Bosnia and Herzegovina | - | 12 | - | 381 | 80 | - | - | 0^a | 1 | - | 4 | 10 | 3 | - | 391 |
| Total EU+ | 6178 | 5935^a | 116911 | 165116 | 34481 | 14010 | 7777 | 13160^a | 15216 | 27168 | 84699 | 111193 | 31177 | 143980 | 276313 |

Area and emissions of EU+ Grassland, Cropland, and Forest Land on peatlands as reported in the NIS 2023 and as worked out in this study. Superscript letters indicated alignment as defined below. For the NIS Forest Land, there are two area values, one for all Forest Land and one for drained Forest Land. From this study, there are also 2 columns, "Forest land" and "Pixel Area", of which the latter includes also forests <0.2 ha. StErr. standard error based on the uncertainty of the IPCC EFs.
aAgreement (± 20%) area or emissions.
bAgreement (±20%) emissions per ha⁻¹.

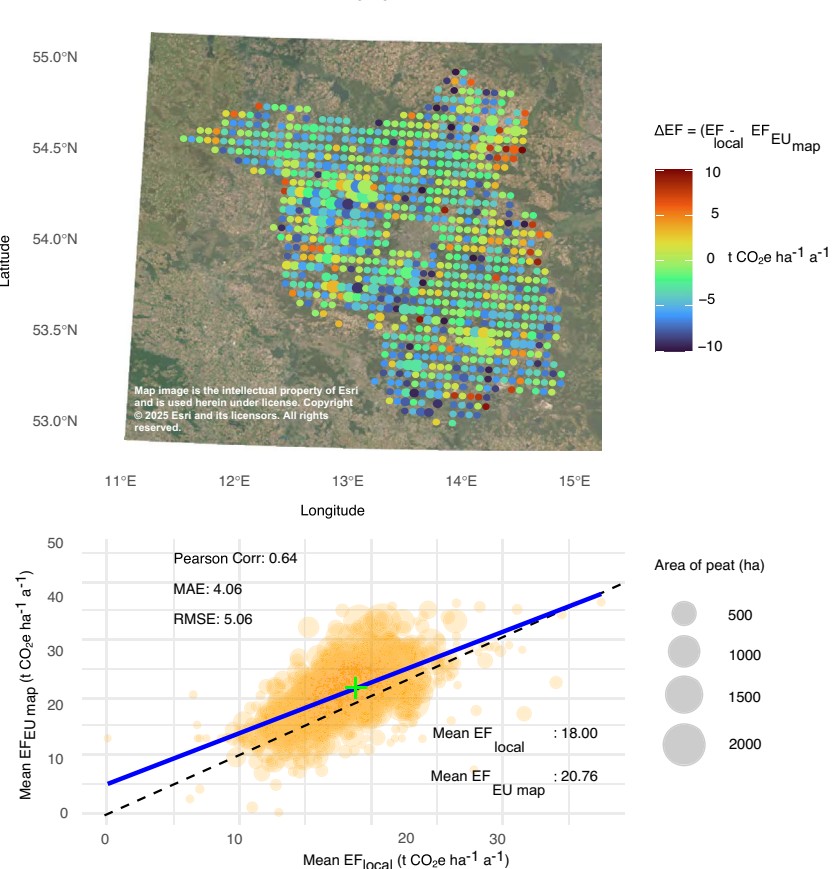

**Fig. 2 | Comparison of EU+ peatland GHG emission map with regional data.** Difference (per 5 km²) between our EU + GHG emission map (Fig. 1B) and a local spatial peatland emission map for the federal state of Brandenburg in Germany (Reichelt, 2021). Red colour indicates lower mean emission in our EU+ map and blue indicates higher estimates in our EU+ map. The size of the dots indicates the cumulative peat area. The point cloud shows the 1 to 1 relation (dashed line) and the regression (blue line) and the emission factor (EF) per pixel, where the size indicates the cumulative peatland area. See Supplement Fig. S2 for more details.

for our substantially higher total GHG emissions from peatlands in EU+ countries compared to their national reporting in NIS (2023) are (1) estimating emissions for all types of land use (Forest Land, Cropland and Grassland) on drained peatlands for all EU+ countries, (2) developing a peatland map from the most up-to-date national datasets (Supplementary Table S.5) and a comprehensive compilation of best available LULC products to estimate the area under each of the land use categories, (3) covering all gases, including $CH_4$ from drainage ditches and DOC losses for all categories and for all countries, and (4) using the appropriate EFs from 2014 IPCC[1] instead of outdated 2006 EFs from IPCC[22].

Many southern and south-eastern EU countries omit emissions from drained peatlands in their reporting. While this may be negligible in several countries with little drained organic soil, Hungary and Romania do and together fail to report about 13 Mt $CO_2$e each year. Other countries seem to underestimate their area of Cropland on drained organic soil. For example, Ireland states that there is only Grassland on agriculturally used peatland[23] (Environment Protection Agency, 2023). Our study, however, suggests that 2.3% of the peatlands in Ireland are used for the production of cereals (9.9 kha) and maize (6.7 kha; Supplementary Table S.1), which causes additional emissions of 0.3–0.7 Mt $CO_2$e depending on the drainage depth class originally assigned to the Grassland.

Our findings also point at under-reporting of the overall agricultural peatland area for Lithuania (having higher national thresholds for SOC in peat and for peat depths, which also is evident for Scotland, UK), for France (having a fragmentary peatland map despite recent

efforts[24]), and for Estonia (Table 1)[15,16]. Many countries may use the EU field registry (Integrated Administration and Control system; IACS) for reporting agricultural areas, but this registry tends to underestimate actual areas. A spatial study comparing the Finnish agricultural area in the IACS and in the National Forest Inventory (used for the GHG inventory) showed that the latter has 12% larger agricultural area because smaller farms, different marginal areas around fields and some ditches are not included in the IACS[25] (Kärkkäinen et al., 2019). Moreover, the inclusion of all relevant gases ($CO_2$, $N_2O$, $CH_4$) and ditch emissions has not been accomplished in all countries yet. For example, peatland emissions of $CH_4$ from drainage ditches (Estonia, Hungary, Lithuania, Netherlands, Poland) and $N_2O$ (partly Poland, Hungary) emissions from peatlands are not reported, or the very low, outdated Tier 1 EF from the IPCC (2006) for Grassland is used (Estonia, Lithuania, Poland). Finally, land use maps can be blind for the differentiating land use intensity of Grasslands, which we derived from the biomass productivity map (see "Methods"). For instance, the UK includes 1278 kha of non-intensive Grasslands in its NIS with very low EFs between −1.0 and 3.3 t $CO_2$e ha⁻¹ a⁻¹, resulting in a low average EF for agricultural peatlands in the UK NIS compared to higher emission estimates when applying Tier 1 default EFs of IPCC (Fig. 1B).

Successful mitigation of GHG emissions from drained peatlands is crucial for achieving EU climate targets by 2050, including climate neutrality[26] and adding 310 Mt $CO_2$ of net sinks within the LULUCF sector[14]. Mitigation measures on drained peatlands necessarily need to include rewetting. Rewetting provides permanent and high emission reductions even though $CH_4$ emissions could increase[27].

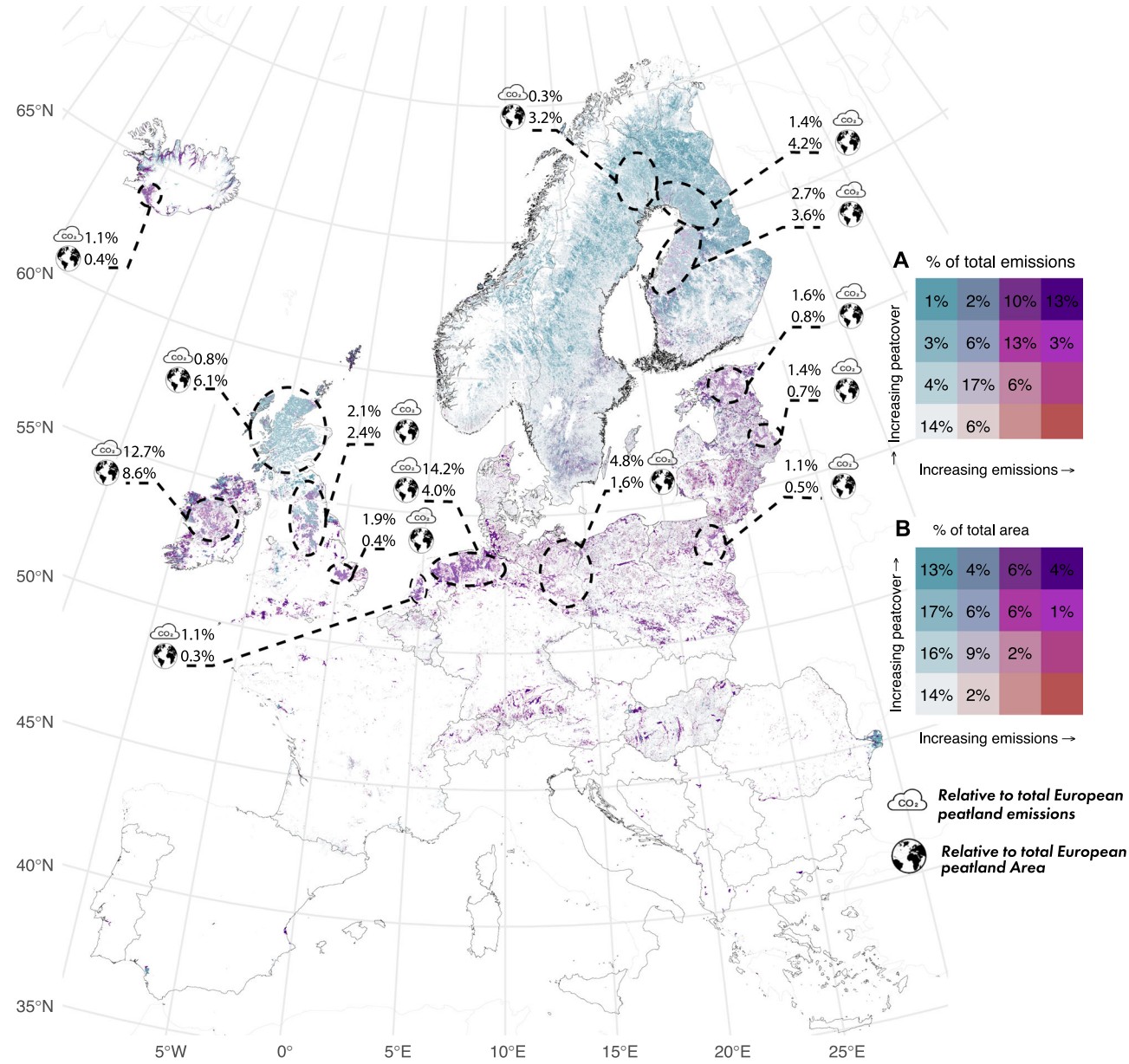

**Fig. 3 | Hotspots of peatland area and GHG emissions in the EU+.** Peatland area and GHG emission ($CO_2$, $CH_4$, and $N_2O$, in $CO_2e$ respectively) density hotspot maps of (drained) peatlands in the EU+, in a 1 km² grid. High peatland density is depicted in blue colours, high emissions intensity in red; the simultaneous presence of a high peatland area and high emissions in purple. Legend matrix **A** (top matrix) shows relative contribution of each color class to total EU+ emissions, and legend matrix **B** (bottom matrix) to total EU+ peatland area.

Rewetting should raise water tables to the surface to fully reduce $CO_2$ emissions, which is necessary to limit global warming to below 1.5 or 2 °C[1]. Mere land use change, while continuing drainage, for example, from agriculture to drainage based forestry maintains considerable emissions[28] (Table 1). The under-reported peatland emissions at the European scale hide the necessity of and blur the potential for their reduction. Countries may be overly optimistic at achieving climate neutrality if emissions are not fully accounted for in national inventories. This study highlights both GHG emission hotspots in Europe and shortcomings in national GHG reporting, which provides a basis for improving national GHG reporting. This may be more easily achieved in the case of Cropland, which is well monitored in the EU. For Forest Land on drained peatland, however, correctly locating the drained areas, mapping their extent and having appropriate EFs in place remains a challenge. This is especially true in countries where agriculture is the dominant type of land use and where awareness of

the role of forestry drained peatlands is lacking. Our analysis provides a methodological approach to include drained Forest Land comprehensibly, which may encourage respective countries to better cover forestry drained peatlands in their GHG inventories.

All currently available EU or European input data has its specific biases, limitations, and uncertainties. We see five potential sources of uncertainty in our input data that may have affected our products: (1) misclassification of specific land use types, (2) neglect of temporary grassland within crop rotation, (3) misclassification of drainage status and nutrient status, (4) uncertainty in EFs, and (5) regionally some under or over-representation of spatial peatland area.

Despite of high accuracies (±70%) of the land use maps, local mismatches do occur[29,30]. Classification errors may arise, for example, when productive Grassland or rangeland vegetation is confused with wetland vegetation due to their similar appearance[31]. Therefore, in areas where Wetlands and Grasslands are adjacent, such as in the

north-western part of Sweden and the far north of Finland, some precaution is required as these areas are most likely undrained Wetlands[32,33]. A similar error may also play a partial role in Grassland classification in Ireland. In addition, some undrained forest seems to be classified as (drained) Forest Land in this study due to insufficient drainage data for forests. This would explain the difference between our Forest Land area and the reported NIR 2023 areas especially for Germany, Estonia, Latvia, Lithuania, Norway, and Poland (Table 1). Yet, our probable overestimation does not produce any notable hotspots on the map. The total high productivity Forest Land area roughly equals the area as reported in NIR 2023 for the countries mentioned and might be preferred as estimate. However, Sweden and Finland report a larger Forest Land area than estimated here, which is an exception from this rule. Area estimates of countries may differ partly because of varying interpretations of whether a wooded peatland is a Wetland or Forest Land.

Another uncertainty factor is the difficulty to distinguish permanent Grassland from short-term Grassland in crop rotation that should be reported as Cropland, using the higher EFs[34]. This is partly resolved by including the 2018 crop map[35] and assigning former Croplands that are now Grasslands to Croplands. However, higher frequency land use maps are needed over a 4–6 years period[34]. Furthermore, the assumption that all shallow drained Grassland is nutrient-rich probably leads to an overestimation of emissions, but in the Wetland Supplement[11] there is no EF for shallow drained nutrient-poor Grassland. This lack of emission data at the time the IPCC EFs were made is related to shallow drained nutrient-poor Grassland not being very common agricultural practice[36]. Some more recent regional assessments deliver insight in emissions from such sites[37,38].

Identification of the drainage level of Grassland using the productivity threshold method provides a reliable approximation (Supplementary Table S.6). A validation of the Forest Land nutrient status based on the productivity map of Tóth et al. (2013) was not conducted[39], although nutrient status does affect emissions. Emissions from boreal Forest Land on drained peatland range from −5.5 up to 20 Mt $CO_2e$ depending on whether the EFs for nutrient-poor or nutrient rich forest are applied. Nutrient status is indeed an important factor in choosing EFs in the Finnish national inventory. Forest types with low nutrient status may have a negative or near-zero soil $CO_2$ balance at present, but as the climate warms, also these forest types are likely turn to net sources of $CO_2$[40]. The lack of validation points for the nutrient levels of forested peatland in the LUCAS soil survey, where only 49 of 21.850 sampling points are from forested peatlands, makes it impossible to use them for validation of EU Forest Land on peatland[35]. New methods to map ditch cover, drainage depth, and estimate drainage impact using remote sensing and machine learning are promising and may yield higher accuracy of drainage extent in European Grassland and Forest Land[41,42].

Using IPCC 2014 Tier 1 EFs provides a standardized and comprehensive approach to GHG reporting and ensures consistency, comparability, and completeness across regions and countries, addressing underestimations caused by outdated data and incomplete reporting emissions of $CO_2$, $CH_4$, and $N_2O$, respectively. However, Tier 1 EFs may not capture the specific conditions of peatlands as accurately as Tier 2 or Tier 3 approaches. A comparison of our maps with two regional spatial emission maps from Germany, based on country- and land-use-specific GHG data, shows that using default IPCC EFs results in somewhat higher average emissions (by 2.8 to 4.0 t $CO_2e$ ha$^{-1}$ a$^{-1}$). Yet, the regional values fall within one standard error of the IPCC factors (Supplementary Fig. S2)[43,44]. If the regional EFs are closer to the actual emissions, our overestimation of temperate Forest Land emissions would be at most 22 Mt $CO_2e$ (−4 t $CO_2e$ ha$^{-1}$ a$^{-1}$ to all temperate Forest Land; the resulting emissions are still more than 3 times higher than reported). We tested the sensitivity of hotspot distribution to variation in EFs in a subset and found

that the overall pattern of hotspots remained similar (Supplement Fig. S.3).

The latest update (2025) of the European Peatland Map (GPD/ Greifswald Mire Centre) is under- or overestimating the peatland area within some countries, when compared to Joosten et al.[15], Barthelmes[16], and Martin & Couwenberg[45]. For Lithuania, we used a data set that includes larger areas of the 'peat in soil' mosaic, whereas the NIS sets high thresholds for C content and peat depth and thus arrives at a smaller area. Also, drained peatland areas are shrinking due to loss of the peat by oxidation under long-term drainage and use. This is expected to accelerate as result of rising temperatures due to climate change[46].

This study provides a map of GHG emission hotspots from peatlands on a 1 km$^2$ grid for EU+ countries. Hotspots emit a disproportionally high amount of peatland GHG emissions in relation to the peatland area, which is linked both to peatland density and to land use intensity. It is crucial to note that the highest cumulative emissions per area do not necessarily imply the highest emission intensity. As our method also considers peatland density, an area where the emissions per hectare are only slightly above average, but where the peatland area is very large, may still be a hotspot. A notable example is the northern region of Ireland, which is characterized by high peatland density and a substantial extent of peatland covered mainly by a mix of drained Grasslands with some drained Forest Land. This combination results in grid cells with emissions that are clearly above average but not as high per hectare as is the case for example, in the north-western part of Germany. It is therefore recommended that the hotspot map is used in conjunction with the GHG emission and land use map (Fig. 1A, B) to develop effective policies and measures to reduce GHG emissions from land use on drained peatlands.

Our results emphasize that GHG emissions from peatlands are not evenly distributed, but that there are regional hotspots across the EU+. Based on our hotspot map, policy incentives can be designed that contribute to reducing these emissions more effectively and efficiently. Our results suggest four regions of major attention: the North Sea hotspot (NW-Germany, the Netherlands, and SE-England), a hotspot region in eastern-Germany, a cluster of hotspots in the Baltic states and eastern-Poland, and two hotspots stretching from Central Ireland to NW-England. Hotspots across these four regions represent 40% of total emissions from EU+ peatlands. Incentives, transformative pathways, and cross-sector policy are needed to efficiently use resources for peatland climate mitigation[47]. Current and future national/EU policy instruments for financing peatland rewetting within e.g., the Common Agricultural Policy, nature restoration plans under the Nature Restoration Regulation (NRR), and carbon farming schemes under the Carbon Removal Certification Framework could direct their funds to regions with highest emission reduction potential. However, aspects such as land prices, stimulating policies, and opportunity costs are also important parameters when prioritising certain regions on the way to achieving the EU climate targets, which ultimately imply the rewetting of all peatlands. Moreover, emissions in other regions of course, need to be curbed as well.

Our results stress the need for country-specific policies that promote mitigation action in specific areas like the Danube Delta in eastern-Romania. Next to addressing the high-emitting peatland of the temperate climate zone, in the boreal zone, mitigation strategies should be considered to reduce emissions from widespread low to mid-emitting peatlands in Forest Land. In general, 20% of all peatland-related emissions come from regions with low peatland cover, which can be advantageous in minimizing conflicts with landowners, because agriculture may not rely on peatlands alone for income[48]. The EU+ hotspot map presented here may be accompanied by detailed national versions in future (emerging from ongoing data collation efforts funded under the Horizon Europe program), which can be instrumental for the national (or trans-boundary) spatial implementation of climate

**Table 2 | Emission factors used in this study**

| Land use category | CO$_2$ | CH$_4$ | N$_2$O | DOC | Ditch | GWP |
|---|---|---|---|---|---|---|
| Boreal | | | | | | |
| Forest Land NP | 0.92 ± 0.88 (n = 59) | 0.33 ± 0.09 (n = 47) | 0.09 ± 0.04 (n = 43) | 0.44 ± 0.1 (n = 10) | 0.15 ± 0.09 (n = 11) | 1.79 ± 0.9 |
| Forest Land NR | 3.41 ± 1.39 (n = 62) | 0.20 ± 0.09 (n = 83) | 1.37 ± 0.6 (n = 75) | 0.44 ± 0.1 (n = 10) | 0.15 ± 0.09 (n = 11) | 5.42 ± 1.72 |
| Cropland | 28.97 ± 5.31 (n = 39) | 0 ± 0 (n = 38) | 5.57 ± 2.42 (n = 36) | 0.44 ± 0.1 (n = 10) | 0.79 ± 0.74 (n = 6) | 35.75 ± 7.51 |
| Grassland | 20.89 ± 10.44 (n = 8) | 0.04 ± 0.02 (n = 12) | 4.06 ± 1.85 (n = 16) | 0.44 ± 0.1 (n = 10) | 0.79 ± 0.74 (n = 6) | 27.03 ± 13.45 |
| Peat extraction | 10.26 ± 5.68 (n = 20) | 0.16 ± 0.13 (n = 15) | 0.13 ± 0.05 (n = 4) | 0.44 ± 0.1 (n = 10) | 0.37 ± 0.37 (n = 6) | 11.36 ± 6.22 |
| Temperate | | | | | | |
| Forest Land | 9.53 ± 2.38 (n = 8) | 0.07 ± 0.04 (n = 13) | 1.14 ± 0.5 (n = 13) | 1.14 ± 0.25 (n = 12) | 0.15 ± 0.09 (n = 11) | 12.93 ± 3.27 |
| Cropland | 28.95 ± 5.31 (n = 39) | 0 ± 0 (n = 38) | 5.57 ± 2.42 (n = 36) | 1.14 ± 0.25 (n = 12) | 0.79 ± 0.74 (n = 6) | 36.45 ± 7.62 |
| Grassland NR, DD | 22.35 ± 4.21 (n = 39) | 0.43 ± 0.41 (n = 44) | 3.51 ± 1.54 (n = 47) | 1.14 ± 0.25 (n = 12) | 0.79 ± 0.74 (n = 6) | 28.22 ± 5.95 |
| Grassland NR, SD | 13.19 ± 1.65 (n = 13) | 1.05 ± 1.13 (n = 16) | 0.69 ± 0.3 (n = 7) | 1.14 ± 0.25 (n = 12) | 0.79 ± 0.74 (n = 6) | 16.42 ± 2.27 |
| Peat extraction | 10.26 ± 5.68 (n = 20) | 0.16 ± 0.13 (n = 15) | 0.13 ± 0.05 (n = 4) | 1.14 ± 0.25 (n = 12) | 0.36 ± 0.32 (n = 6) | 12.05 ± 6.5 |

*n* number of sites used for the estimate, *NP* nutrient poor, *NR* nutrient rich, *DD* deep drainage, *SD* shallow drainage, GWP = 100-year global warming potential (IPCC, AR6), has been used and including the uncertainty within each GWP conversion.
Emission factors (t CO$_2$e ha$^{-1}$ annually$^{-1}$ ± standard error of each EF and number of sites *n*) for CO$_2$, CH$_4$, N$_2$O, DOC, and ditch emissions by land use category and climate zone. Source: IPCC (2014).

mitigation action in peatland-rich areas[47]. The accuracy of our hotspot map would benefit from additional ground-truthing, for example, in Eastern Poland, Central Finland, and Estonia. Nevertheless, all these regions are likely to need targeted mitigation measures. Peatland climate mitigation action on the continental scale would benefit from a dialogue and concerted efforts together with peatland-rich countries neighboring eastern-Poland (Ukraine and Belarus).

Several European policy initiatives encourage to rewet peatlands (e.g., thex NRR[49]), but instruments are lacking for developing EU-wide and national policies to target restoration efforts[50], notably data on extent, condition, and related GHG emissions of peatlands[51]. Our findings emphasize the scale of climate mitigation benefits achievable through targeted rewetting of (agriculturally used) peatlands in emission hotspots. Since agriculturally used peatlands cover only 3% of the agricultural land in the EU and the EU is a net food exporter, this would not have much relevance for food security[52], certainly when seen in the light of necessary shifts in diet and resource efficiency[53]. Our land use and GHG hotspot maps are intended to close knowledge gaps and fine-tune the development and implementation of peatland restoration policies and action across the EU+. Furthermore, the hotspot map can be used on a (sub-)national level, and across national borders to tackle emissions hotspots on a transboundary level, similar as in biodiversity conservation[54].

Despite inherent uncertainties in peatland emission mapping, opportunities for substantial climate mitigation exist. Immediate action and mitigation policy focussing on identified hotspots and peat-rich areas should be central. Full peatland restoration and rewetting offer the most effective climate mitigation by halting emissions and creating carbon sinks. To meet Paris Agreement targets, rapid water table raising is crucial, necessitating the development of paludiculture with water-adapted crops and trees as a European and global key priority.

## Methods

### Land use map for peatlands

Spatial peat (and peaty) soil data for EU+ countries (EU member states plus Albania, Andorra, Bosnia and Herzegovina, Iceland, Liechtenstein, Montenegro, North Macedonia, Norway, Serbia, Switzerland, United Kingdom) were taken from the peatland layer of the European Wetland Map[55]. The dataset was compiled from over 100 different national and regional datasets. It includes various geodata (raster, polygon, and point data), mainly from publications on peatland and soil research, national or regional soil databases, OpenStreetMap, and partly from national or regional data provided by research institutes, ministries, or

authorities[55]. Their definitions of peatlands are in line with the 2006 and 2014 IPCC definition of 'organic soils.

The EU crop map developed by the Joint Research Centre (JRC), based on satellite images of 2022 with a spatial resolution of 10 m was used to stratify the peatland map according to the 2022 land cover distribution. This map delineates the most common (19) crops grown on agricultural parcels in the EU with an overall accuracy of 71%[29]. However, it does not include non-agricultural areas and does not detect grasslands within crop rotation. Nor does it include countries outside the EU. Therefore, the land use map of Witjes et al.[30] was used as a base layer to distinguish land use beside the agricultural areas. This land use map has a lower resolution of 30 m and even though it has 40 different land use classes, it distinguishes only three types of agricultural land (non-irrigated, irrigated, and grassland). So it was only used to fill the gaps in the JRC map. Furthermore, we included the EU crop map of d'Andrimont et al.[35], which formed the basis for the Ghassemi et al.[29] map, but is based on satellite images of 2018 to detect grasslands within crop rotation. We assume that the land cover over the years has not changed much, as, e.g., the difference between the 2020 and 2021 *World cover* map developed by Zanaga et al.[56] and long-term land use changes in the EU are small[57]. Area was calculated in hectares per grid cell. For our analysis, the 80 single land use types were aggregated into five main classes: Grassland, Cropland, Forest Land, Wetlands, peat extraction, and Build-up (Supplementary Table S.1). A forested area is in national inventory reporting counted as Forest Land if at least 90% of a 0.25 ha area is classified as forest[58]. We also included smaller forested areas in the Pixel Area estimates, whereas for the Forest Land we adhered to the official definition (Table 1). Furthermore, Tegetmeyer et al.[55] provided detailed information about peat types in certain areas, indicating near-natural conditions with classifications such as "blanket bog" and "heather moor". If the land use map categorised these areas as Grassland, they were reclassified as Wetland.

### Emission factors for GHG emissions from peatlands

Drainage influences the three most important GHGs emitted from peatlands, CO$_2$, CH$_4$, and N$_2$O, in such a way that less drainage results in lower overall GHG emissions. Drained peatlands under higher temperatures (e.g., in temperate zones) emit more GHGs than peatlands under lower temperatures (e.g., in boreal zones) and nutrient-rich peatlands emit more GHGs than nutrient-poor ones[11]. Default IPCC EFs are stratified per climate zone and land use class (Table 2). Peatland drainage requires ditches, so the emissions from these ditches must be included as well. Due to limited EU-wide data, default IPCC fractions of

5% ± 2.5 for agricultural fields and 2.5% ± 1.25 for Forest Land were applied[11]. We did not include GHG emissions from (wild)fires.

## Allocation of nutrient and drainage levels

To apply appropriate EFs (IPCC, 2014, chapter 2.2) to the land use map of peatlands, we had to allocate nutrient and drainage levels. As there is currently no suitable European-wide map of nutrient and drainage levels in peatlands, the biomass productivity map of Tóth et al. ($R^2$ of 0.85)[39] was used as a proxy. We assumed that more productivity means a higher probability of drainage and fertilizer application. Thus, we classified a Grassland area "deep drained" if it has "high biomass production" or "shallow drained" if it has "low biomass production" according to Tóth et al.[39] map. The same was done for Forest Land, which was classified "nutrient rich" with "high biomass production" or "nutrient poor" with "low biomass production" according to Tóth et al.[39] map. Furthermore, we assumed that biomass productivity has not changed considerably between 2013 and 2022 as nutrient content in peat soils does not change in the short term—even with apparent changes in management[58,59].

In order to apply the biomass productivity map in this way, two threshold values had to be implemented for the biomass production level distinguishing, (1) whether a Grassland has a high or low agricultural activity; and (2) whether a Forest Land is nutrient-rich or nutrient-poor (see Supplementary Table S.6). To identify the threshold value, we applied the mean value and the minus one standard deviation as two potential threshold values per country and per land use type and checked whether the ratio between "deep drained" and "shallow drained" follows the given values in the (updated) 2020 UNFCCC National Inventory Submission[16]. This was achieved by overlaying the peatland land use maps per country with the biomass productivity map using ArcGIS Pro 2.9. When applying the mean Grassland Productivity Index (PI) of 6.32 across EU to distinguish if a peatland Grassland is shallow drained (below 6.32 PI) or deep drained (above 6.32 PI), 8 out of 13 countries were estimated to agree well (within a ± 5% range) with a previous estimate of the distribution of deep/shallow drained Grassland[16] (Supplementary Table S.6). As there was no European-wide information available about the deviation of nutrient-rich and nutrient-poor Forest Land and wetlands on peatland the mean PI of 6.08 was used as the threshold value for Forest Land.

## Peatland GHG emission map and GHG emission hotspot map

With a resolution of approximately 10 meters peatland emissions were estimated using the EFs of IPCC (2014), the land use map, the Köppen-Geiger climate classification (period 1990–2020 produced by Beck et al., 2023) and the productivity map of Tóth et al. (2013)[39] with corresponding best fit thresholds (see above) to assess the nutrient level of Forest Land and the drainage level of Grassland (Supplement Fig. S1 Dichotomous decision tree diagram). The emissions (t $CO_2$e) were calculated for each grid cell by using R-4.4.1[60]. In order to assess whether a peatland area is a GHG hotspot or not, we used the "biscale" package[61] in R. This package uses thematic choropleth mapping with two variables, in this case peatland density (amount of peatland per area) and cumulative peatland GHG emissions per area. The peatland density and the summed GHG emissions within each certain area were calculated on a 1 km resolution. This resolution gave a good visual representation of where the peatland emission hotspots are located and highlights the highest emitting and the densest peatland areas at the chosen EU+ scale. The highlighted selected areas in Fig. 3 were based on a lower resolution 10 km hotspot map. The "fisher" option that works best with skewed data has been used within the "biscale" packages, as the emissions are not normally distributed, due to predetermined EFs.

## Validation

After developing the GHG emission (hotspot) map, an overview table on the national level emissions was developed for validation. Here,

emission estimates of the 2023 NIS (UNFCCC 2023) were compared to the outcomes of our study. The relevant peatland area and emission values were aggregated from the CRF categories 3D, 4A-D, 4II, and the files of the 2023 NIS. Furthermore, the aggregated EU + LULC map from this study was validated using two regional GHG emission maps for peatlands in Germany, namely for Brandenburg[43] and Mecklenburg–Vorpommern[44]. Both maps use different EFs compared to each other and to our study. This evaluation was done by overlaying the map and comparing the mean emissions within sub-regions of around 5 km$^2$.

## Inclusion & ethics statement

This research aligns with the Inclusion & Ethical guidelines embraced by Nature Communications.

## Reporting summary

Further information on research design is available in the Nature Portfolio Reporting Summary linked to this article.

## Data availability

The land use and greenhouse gas emission data generated in this study have been deposited in the Zenodo database available on https://zenodo.org/records/17091811. The raw data used to derive these maps are described in the "Methods" section and originate from previously published datasets; these are not provided here but can be accessed through the original publications cited in the manuscript.

## Code availability

The scripts that support the findings of this study are available on https://zenodo.org/records/14974022.

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

## Acknowledgements

This research was funded through the 2019–2020 BiodivERsA joint call for research proposals, under the BiodivClim ERA-Net COFUND programme (PRINCESS project), with the funding organisations Research Council of Finland and the Federal Ministry of Education and Research (BMBF) through VDI-VDE (Germany). Part of the research was funded through PaluWise (Horizon Europe GAP-101181479), FIBSUN (GAP-101112318), and by Deutsche Forschungsgemeinschaft (DFG, German Research Foundation)—SFB Transregio Collaborative Research Centre 410/1 2025-531801029 "WETSCAPES2.0". C.F. was supported by Wet Horizons (Horizon Europe GAP-101056848).

## Author contributions

Q.v.G., K.L. and F.T. conceptualized and designed the research. Q.v.G., A.B., F.T. and J.C. developed the methodology. Q.v.G., A.B., F.T. and C.F. validated the findings. Q.v.G. was responsible for the visualizations. Formal analysis was performed by Q.v.G. and partly by A.B. Data curation was conducted by both Q.v.G., A.B. and C.T. The original draft was prepared by Q.v.G., F.T., K.L., A.B., C.F., J.C. and N.M. F.T., C.F., A.B. and K.L. acquired funding. All authors contributed to the review and editing of the manuscript.

## Funding

## Competing interests

The authors declare no competing interests.
