## [Transparent Peer Review file · Nature Communications]

Identifying hotspots of greenhouse gas emissions from drained peatlands in the European Union

Corresponding Author: Dr Franziska Tanneberger

Version 0:

Reviewer comments:

Reviewer #1

(Remarks to the Author)

Summary

The paper titled "Identifying hotspots of greenhouse gas emissions from drained peatlands in the European Union" offers an in-depth analysis of greenhouse gas (GHG) emissions from peatlands, presenting a detailed mapping of emission hotspots across the EU. The study highlights significant underreporting in national inventories and identifies key regions for targeted mitigation efforts.

Strengths

- The paper provides a comprehensive and detailed mapping of peatland GHG emissions across the EU, utilizing high-resolution spatial datasets.
- The identification of specific emission hotspots is particularly valuable for guiding targeted mitigation efforts.
- The study addresses significant gaps in current national GHG inventories, offering practical suggestions for improving reporting accuracy.
- The methodology is robust, integrating various data sources to enhance the accuracy of emission estimates.

Weaknesses

- The use of datasets from different time periods and countries may introduce inconsistencies.
- Emission factors based on default IPCC values might not fully capture regional variability.
- There are some underestimations in the forest land area, particularly in Finland and Sweden, which could impact overall emission estimates.

Minor Comments

- Simplify some of the more complex sentences to enhance readability.
- Add more explanatory notes to figures and tables for clarity.
- Clearly outline all assumptions made in the methodology, particularly regarding biomass productivity and drainage classification.

Recommendation

Overall, the paper presents significant findings that contribute to our understanding of GHG emissions from EU peatlands and the necessary steps for mitigation. However, addressing the identified weaknesses will further strengthen the study. I recommend the paper for major revision.

Detailed Revision Recommendations

1. Methodology Improvements

- Data Consistency:
 - o Ensure all datasets used are clearly documented with their time periods. Acknowledge any potential inconsistencies due to differing data periods and discuss their possible impact on the results in a dedicated section.
- Emission Factors:
 - o Justify the use of default IPCC emission factors by comparing them with any available region-specific factors. If region-specific factors are not used, explain why and discuss any limitations this might introduce.
- Nutrient Status Validation:

o Include a brief explanation of the limitations due to the lack of validation for the nutrient status of forested peatlands. Suggest potential future studies or data sources that could address this gap.

- Drainage Level Classification:

o Detail the assumptions made for classifying drainage levels and consider using additional proxy indicators if possible. Discuss the limitations and possible errors due to these assumptions.

2. Data and Analysis

- Land Use Intensity:

o Ensure the classification of land use intensity is clear. If possible, incorporate additional data or proxies to better differentiate between intensive and non-intensive agricultural practices.

3. Reporting and Discussion

- Detailed Assumptions:

o Clearly outline all major assumptions made in the methodology section, particularly those regarding biomass productivity and drainage classification. Discuss their potential impact on the results in a limitations section.

- Uncertainty Analysis:

o Provide a brief uncertainty analysis that highlights the major sources of potential error and their expected impact on the study's conclusions in the discussion section. Use simple sensitivity analyses to illustrate these points if possible.

- Comparative Analysis:

o Compare the study's emission estimates with those from national inventory submissions and discuss any significant differences. Highlight areas where the study's methods provide improved accuracy.

- Policy Implications:

o Expand the discussion on policy implications to include specific recommendations for policymakers based on the identified hotspots. Emphasize the importance of targeted mitigation measures in these areas.

Reviewer #2

(Remarks to the Author)

Comments on Giersbergen et al. Nature Communications

This paper attempts to identify drained wetland GHG hot spots for the EU. The new component is the overlaying of drained wetland data and agricultural land-use. This makes the areas involved different than previous analyses. However, the authors use the emission factors from Wilson et al., and these have huge uncertainties. The authors discuss the uncertainties in their land-use data set but accept the EFs without question? There authors make attributions that are not correct and use terms that are not relevant. With some work this could be a useful addition, but I not sure Nature Communications is the right place.

Comments

Ln 11 The only thing that makes these emission hot spots is the attribution of EFs to a specific land-use drained wetland class that appears in these locations. There is nothing special about the EFs - they are Wilson's update of the IPCC's update. These has been shown to have a seasonal bias to them by He and Roulet. Since the seasonality is largely latitude-dependent, what implication does this bias have to this work?

Ln 84 McCain et al. never suggested this. They did not mention peatlands in the 2003 article. Maybe the sentence is poorly written, but the attribution is wrong. McCain and the idea of HSHM is fine.

Ln 151 and throughout Tipping pt? Why is this a tipping point? Table S2 is productivity by country. What are the forcing variables for a tipping point?

Ln 184 Evaluating your outcome against the methodology used to develop it? Is it an independent evaluation? More details are needed here as this will allow the reader to determine if the results are a significant addition or not.

Since you are using fixed EFs from others, the only thing being evaluated here is the land-use wetland drainage maps, which are correct. Are there no independent maps from other analyses one could use for evaluation? I am thinking of the detailed work done by Connolly in Ireland.

Ln 194-202 I Don't see why percentage matters here, as the countries are not the same size? It is the absolute areas in each country that contribute emissions. Using percentages is misleading. Ireland and Scandinavia have a higher percentage of undisturbed peatlands, but they both have a significant area of drained peatlands that are emitting GHGs. Sweden is the second or third largest country in Europe (I recognize Europe and the EU are not the same, but you have Britain listed in this paragraph), depending on whether you consider the Ukraine part of Europe. Conversely, some countries on your list are tiny, so their contribution is small.

Ln 258 This is an interesting map. The map results are primarily from drained peatlands in crop and grassland, as driven by the EFs assigned to those two classes. There is a lot of uncertainty in the EFs. Did you do any sensitivity analysis using the range of EF values in each category?

Ln 279 Not a huge number when overall emissions are considered. Also, you compare this with other sources, and one can assume these are equally uncertain. Emissions reporting is a bit of a game.

Ln 364-368 In the previous section, you were referring to 2050. However, this argument is based on the GWPs of rewetted replacing drained using 100-year GWPs. 2050 is 25 years away so the 20-year GWP would be more honest. The picture would look very different if you use 20-year GWPs. The methane component becomes much more critical. I think you are beginning to mix apples and oranges here by discussing EU 2050 climate policy and a 100-year GWP. I don't think anyone can seriously argue that peatland restoration and rewetting are not good ideas in the long run, but you have hooked your argument to EU climate policy, so stick to the relevant time horizons for the policy. The 20-year GWPs can be figured out just as quickly as the 100-year.

Ln 376 No uncertainty in the attribution and magnitude of the EFs?

Reviewer #3

(Remarks to the Author)

The major claim of the paper is providing a detailed maps of EU peatland use and its associated GHGs, and current NIR to UNFCCC underreported the emissions. The paper presents a literature synthesis for both mapping and emission estimation although the methods both for land use classification and emission factors (EF) can not be deemed as novel.

Whereas various emission estimates for EU have been made before (earlier ones e.g. Byrne et al 2004 EU peatlands: Current Carbon Stocks and Trace Gas Fluxes), this study updates those estimations. One improvement to further classify the land use categories was made through the allocation of nutrient and drainage levels by biomass productivity. This has to be made due to lack of data, but my concern is it differs from the IPCC EF classifications (by minerogenic vs ombrogenic and mean water table depth of 30 cm) increasing uncertainties simply due to two classifications are used for the area and EF. Would this influence the EFs and the GHG budgets? To which extend we should further classify the land use without introducing extra uncertainty?

The EFs were directly taken from Wilson et al. 2016. There are more recent updates of EF for subcategories, for instance, for forests on organic soils e.g. Jauhiainen et al 2023 revised the EF and should be used to update the respective EFs.

I would further strongly suggest the authors to have a clearly defined boundary for their fluxes in this study i.e. to define what fluxes are included in their emission estimations. Would emissions from aquatic fluxes included? One aspect that is totally ignored fire emission. Drained peatlands are known to have fire probability and with climate change the risk is higher, which can cause events-based C fluxes. Fire emissions are included in the equations of Wilson et al 2016 for completion but no data were really incorporated in those EFs.

References

Christensen TR, Friberg T (lead authors) with Byrne KA, Chojnicki B, Drösler M, Freibauer A, Froking S, Lindroth A, Mailhammer J, Malmer N, Selin S, Turunen J, Valentini R, Zetterberg L, Vandewalle M. 2004. EU peatlands: Current carbon stocks and trace gas fluxes, Report 4/2004 to 'Concerted action: Synthesis of the European Greenhouse Gas Budget', Geosphere-Biosphere Centre, Univ. of Lund, Sweden

Jauhiainen, J., Heikkinen, J., Clarke, N., He, H., Dalsgaard, L., Minkkinen, K., Ojanen, P., Vesterdal, L., Alm, J., Butlers, A., Callesen, I., Jordan, S., Lohila, A., Mander, Ü., Óskarsson, H., Sigurdsson, B. D., Søgaard, G., Soosaar, K., Kasimir, Å., Bjarnadottir, B., Lazdins, A., and Laiho, R.: Reviews and syntheses: Greenhouse gas emissions from drained organic forest soils – synthesizing data for site-specific emission factors for boreal and cool temperate regions, *Biogeosciences*, 20, 4819–4839, <https://doi.org/10.5194/bg-20-4819-2023>, 2023.

Version 1:

Reviewer comments:

Reviewer #1

(Remarks to the Author)

The authors have addressed all my reviews, and I would suggest this paper could be published in Nature Communications.

Thanks.

(Remarks on code availability)

Reviewer #3

(Remarks to the Author)

The authors have done a really good job in addressing my comments. I also believe that the sensitivity analysis with EFs and the benchmarking with regional estimates improve the paper. I have no more comments for the authors at this stage.

(Remarks on code availability)

Reviewers Comments and Response (in blue)

Reviewer #1 (Remarks to the Author):

Summary

The paper titled "Identifying hotspots of greenhouse gas emissions from drained peatlands in the European Union" offers an in-depth analysis of greenhouse gas (GHG) emissions from peatlands, presenting a detailed mapping of emission hotspots across the EU. The study highlights significant underreporting in national inventories and identifies key regions for targeted mitigation efforts.

We sincerely thank the reviewers for their thorough and constructive evaluation of our manuscript. We appreciate the recognition of the strengths of our study and our approach to identifying key regions for targeted GHG mitigation efforts. The reviewer's feedback has been enormously valuable in guiding our revision and improving the manuscript.

We have carefully considered each of the reviewer's points and made significant adjustments to address the concerns raised. The comments on data consistency, emission factors, and methodology were particularly helpful, and we have worked to ensure that these aspects are now clearly explained and justified in the revised manuscript. We hope that our responses and updates comprehensively and effectively address all of the reviewer's concerns.

Strengths

The paper provides a comprehensive and detailed mapping of peatland GHG emissions across the EU, utilizing high-resolution spatial datasets.

The identification of specific emission hotspots is particularly valuable for guiding targeted mitigation efforts.

The study addresses significant gaps in current national GHG inventories, offering practical suggestions for improving reporting accuracy.

The methodology is robust, integrating various data sources to enhance the accuracy of emission estimates.

Weaknesses

The use of datasets from different time periods and countries may introduce inconsistencies.

Thank you very much for the feedback. We agree that using datasets from different time periods is not ideal. Since we compare our results to the GHG emission data for the year 2021 reported in the National Inventory Submission (NIS) of 2023, which reports emissions for 2021, and our primary land use dataset (crop map) originates from satellite imagery from 2018, there is a potential mismatch due to crop changes. Even though we mentioned in the original manuscript that we found minor changes in area of different land uses it may introduce inconsistencies.

During the revision of the paper, a new crop map based on 2022 satellite imagery¹ became available, which we have now used. Additionally, we incorporated the latest peatland distribution maps recently (end of 2024) updated by the Global Peatland Database, resulting in increased peatland extent, particularly in the Balkans and central eastern European regions.

Regarding land productivity, short-term land use changes do not significantly affect soil nutrient content². Therefore, we do not expect major changes in productivity even if land use changes between years. Furthermore, the extent of rewetting in recent years has been minimal³ and is unlikely to impact land use productivity on an EU scale.

We therefore added added in line 151-153 of the revised manuscript: “*Furthermore, we assumed that biomass productivity has not changed significantly between 2013 and 2022 as nutrient content in peat soils does not change in the short term - even with apparent changes in management (Heuts et al., 2024)*”

Emission factors based on default IPCC values might not fully capture regional variability.

We agree, this is an important point. To address this, we added a section in the revised manuscript (line 264-277 of the revised manuscript) comparing regional spatial peatland GHG emission maps from two German federal states representing peatland-rich regions within our emissions map. Both regional maps^{4,5} were developed using emission factors different to those in our study and therefore serve as indicators for evaluating our estimates on a regional scale. Also in the discussion this comes back in lines 406-416 of the revised manuscript.

Based on this assessment, the cumulative emissions per region fall within the 1 standard error of the IPCC emission factor estimates in our emission map. Regional estimates are often slightly lower (2-4 t CO₂e ha⁻¹ a⁻¹) than the mean IPCC emission factor, particularly for lower-emitting land uses categories such as Forest Land. However, the overall values remain in reasonable agreement.

We also assessed the potential impact of reducing Forest Land emissions by 4 t CO₂e ha⁻¹ a⁻¹. Even with this adjustment, the emissions remain more than three times higher than those reported in the National Inventory Submissions (NIS) 2023. Therefore, we are confident that our map adequately captures regional variability.

1 Ghassemi, B., Izquierdo-Verdiguier, E., Verhegghen, A., Yordanov, M., Lemoine, G., Moreno Martínez, Á., ... d'Andrimont, R. (2024). European Union crop map 2022: Earth observation's 10-meter dive into Europe's crop tapestry. *Scientific Data*, 11(1), 1048. doi:10.1038/s41597-024-03884-y

2 Heuts, T. S., van Giersbergen, Q., Nouta, R., Nijman, T. P. A., Aben, R. C. H., Scheer, O. van der, ... Fritz, C. (2024). Shallow drainage of agricultural peatlands without land-use change: have your peat and eat it too. *Frontiers in Environmental Science*, 12. doi:10.3389/fenvs.2024.1437394

3 Andersen, R., Farrell, C., Graf, M., Muller, F., Calvar, E., Frankard, P., ... Anderson, P. (2017). An overview of the progress and challenges of peatland restoration in Western Europe. *Restoration Ecology*, 25(2), 271–282. doi:10.1111/rec.12415

4 Reichelt, F. (2021). Treibhausgas-Emissionen aus organischen Böden in Brandenburg. Greifswald Moor Centrum-Schriftenreihe 02/2021 (Selbstverlag, ISSN 2627- 910X), 11 S

5 Ministerium für Klimaschutz, Landwirtschaft, ländliche Räume und Umwelt Mecklenburg-Vorpommern. (2025). Strategie zum Schutz und zur Nutzung der Moore in Mecklenburg-Vorpommern. Schwerin, Germany. Retrieved from <https://www.regierung-mv.de/serviceassistent/download?id=1675720>

There are some underestimations in the forest land area, particularly in Finland and Sweden, which could impact overall emission estimates.

We agree that the Forest Land area was underestimated in the first submission, particularly in Finland and Sweden. This issue was due to methodological choices when classifying Forest Land versus Wetland. Initially, areas classified as forested Wetland should have been categorized as Forest Land under UNFCCC guidelines. We have now adapted the classification to ensure that undrained Forest Land remains categorised as Forest Land, bringing the estimates more in line.

Additionally, there was overestimation in the Forest Land area besides Finland and Sweden. This was due to the presence of trees along built-up areas (e.g., roads, railways, and buildings) that should not be classified as Forest Land according to FAO/UNFCCC guidelines. FAO defines Forest Land as: "Forest is a minimum area of land of 0.05-1.0 hectares with tree crown cover (or equivalent stocking level) of more than 10-30 percent, with trees capable of reaching a minimum height of 2-5 meters at maturity in situ."

IPCC defines Wetlands as: "Land covered or saturated by water for all or part of the year (e.g., peatlands) that does not fit into the forest, cropland, grassland, or settlement categories defined in Section 2.2 of this report."

This means that undrained Forest Land is still a Forest Land according to the IPCC 2006 reporting guidelines. However, a more suitable approach for reporting emissions from undrained Forest Land would be the classification criteria from Cabbage and Flather (1993)⁶, which consider vegetation, soils, and hydrology.

- 1) Wetland vegetation, categorized into obligate hydrophytes (present >99% of the time), facultative wetland species (67%-99% probability), and facultative species (33%-67% probability)⁷
- 2) Wetland soils are characterized by high organic matter content, slow decomposition, mottled gray appearance, and anaerobic conditions.⁷
- 3) Hydrology involves prolonged flooding or water saturation.

However, since this approach does not align with the IPCC guidelines used in the NIS and as it is difficult to apply using remote sensing data, it was not adopted.⁷

Minor Comments

- Simplify some of the more complex sentences to enhance readability.

We thank the reviewer for pointing this out and agree that less complex sentences increase readability. We have simplified sentences throughout the manuscript.

- Add more explanatory notes to figures and tables for clarity.

We have included extra explanations in the revised figures and tables, and also within the text.

⁶ Cabbage, F. W., & Flather, C. H. (1993). Forested Wetland Area and Trends. *Journal of Forestry*, 91, 48-54.

⁷ COWARDIN, L.M., V. CARTER, EC. GOLET, and E.T. LAROE. 1979. Classification of wetlands and deepwater habitats of the United States USDI Fish & Wildl. Serv., Washington, De Publ. FWS/OBS-79/31. 103 p.

Clearly outline all assumptions made in the methodology, particularly regarding biomass productivity and drainage classification.

In the original manuscript, the assumption regarding land use maps across different periods was mentioned in method section “Land use map for peatlands” and for the productivity map in the method section “Allocation of nutrient and drainage levels”.

As there is already a limitation and uncertainty section in the discussion, now expanded with a discussion on the use of IPCC emission factors and a comparison with regional maps, we agree that the methodology lacked a clear overview. This made it difficult for readers to identify key uncertainties. To address this, we have created a detailed methodology overview, which is now included in the supplementary materials (Figure S1). We preferred not to place it in the main text to keep the reader's focus on the primary findings and to meet the space requirements.

Additionally, we assumed that biomass productivity remained largely stable between 2013 and 2022, as the nutrient content in the soil does not change rapidly even with significant management changes⁸. This has been included in the revised manuscript and addressed this concern already in an earlier reply (see above).

Recommendation

Overall, the paper presents significant findings that contribute to our understanding of GHG emissions from EU peatlands and the necessary steps for mitigation. However, addressing the identified weaknesses will further strengthen the study. I recommend the paper for major revision.

Detailed Revision Recommendations

1. Methodology Improvements

- Data Consistency:

- o Ensure all datasets used are clearly documented with their time periods.

We addressed this recommendation already in an earlier reply (see above), explaining the updates made to the peatland distribution and land use maps in order to improve clarity and consistency. Additionally, we have added a methodology overview figure with time periods.

Acknowledge any potential inconsistencies due to differing data periods and discuss their possible impact on the results in a dedicated section.

We followed the reviewer's suggestion and addressed this concern already in an earlier reply (see above), where we explained the updates made to the peatland distribution maps and land use map to enhance clarity and consistency and added a methodology overview figure.

⁸ Heuts, T. S., van Giersbergen, Q., Nouta, R., Nijman, T. P. A., Aben, R. C. H., Scheer, O. van der, ... Fritz, C. (2024). Shallow drainage of agricultural peatlands without land-use change: have your peat and eat it too. *Frontiers in Environmental Science*, 12. doi:10.3389/fenvs.2024.1437394

- Emission Factors:

- o Justify the use of default IPCC emission factors by comparing them with any available region-specific factors. If region-specific factors are not used, explain why and discuss any limitations this might introduce.

We seized the reviewer's recommendation and have added a comparison using region-specific emission factors. We addressed this point already in an earlier reply (see above) and summarize it here for the sake of clarity. A new section has been added to the revised manuscript comparing our emissions map, based on default IPCC emission factors, with regional spatial emission maps from two federal states in Germany^{9,10} that use region-specific factors. This comparison provides an indicator of the robustness of our approach on a regional scale. Our assessment shows that the cumulative emissions per sub-region within each of the regional maps derived from our map generally fall within one standard error of the IPCC emission factor estimates. While the regional maps often yield slightly lower emissions than our mean values, the results are overall consistent. This alignment gives us confidence that our emissions map is suitable for assessing regional variability despite using the default IPCC emission factors. We acknowledge, however, that region-specific factors might capture localized conditions more precisely. This limitation has been noted and discussed in the revised manuscript to provide a balanced perspective on the implications of our approach.

Jauhiainen et al., 2023⁹ found minor differences with the more detailed updated values for Forest Land, so we would expect that the 22 Mt CO_{2e} is really an maximum overestimation.

In line 471-484 (within the uncertainty and limitations section) of the revised manuscript the necessary adaptations had been done related to this comment.

- Nutrient Status Validation:

- o Include a brief explanation of the limitations due to the lack of validation for the nutrient status of forested peatlands. Suggest potential future studies or data sources that could address this gap.

Many thanks for pointing at the challenges of nutrient status validation. In the peatland distribution map, there is information regarding the peatland type. However, this information is not available for the majority of peatlands in the EU+, and clearly not for entire countries. Adding to this challenge a naturally nutrient-poor bog peatland can be currently nutrient-rich, due to land use (e.g. fertilization) or prolonged periods of drainage. Therefore, only for specific peatland types — such as "blanket bog," "raised bog," "oligotrophic fen complexes," "oligotrophic," "heather moor," "montane vegetation," "poor fen," "quaking bog," and "condensation mire" — we assigned nutrient-poor boreal forest or shallow-drained temperate grassland peatland types. For other peatland types, we continued using the productivity map for nutrient status classification of Forest Land, although we lacked a tool to calibrate the threshold value. This has been addressed in the revised manuscript method section in line 123-126. We additionally also included sensitivity analyses of the nutrient status classification in forested peatlands based on comments from other reviewers 2 and 3 (lines 391-393 of the revised manuscript).

⁹ Jauhiainen, J., Heikkinen, J., Clarke, N., He, H., Dalsgaard, L., Minkinen, K., Ojanen, P., Vesterdal, L., Alm, J., Butlers, A., Callesen, I., Jordan, S., Lohila, A., Mander, Ü., Óskarsson, H., Sigurdsson, B. D., Sogaard, G., Soosaar, K., Kasimir, Å., . . . Laiho, R. (2023). Reviews and syntheses: Greenhouse gas emissions from drained organic forest soils – synthesizing data for site-specific emission factors for boreal and cool temperate regions. *Biogeosciences*, 20(23), 4819–4839. <https://doi.org/10.5194/bg-20-4819-2023>

- Drainage Level Classification:

- o Detail the assumptions made for classifying drainage levels and consider using additional proxy indicators if possible. Discuss the limitations and possible errors due to these assumptions.

We attempted to use the drainage map from OpenStreetMap; however, it proved unreliable because it only includes drains of significant size. Narrow ditches are missing, and relying solely on the visible larger ditches could mislead readers into thinking all ditches were included to assess drainage, despite our clarification that only major ditches were included. Furthermore, it is unclear whether wide ditches are deep or shallow. And in addition, the presence of a ditch alone does not provide information about the drainage level, as ditches may also serve for irrigation and water supply. Therefore, we continued using biomass productivity while overriding it with peatland type information when available, as discussed in the previous comment.

2. Data and Analysis

- Land Use Intensity:

- o Ensure the classification of land use intensity is clear. If possible, incorporate additional data or proxies to better differentiate between intensive and non-intensive agricultural practices.

Please see above the changes made in the land use classification and estimation of productivity, the related method descriptions and the new section on uncertainties and limitations.

3. Reporting and Discussion

- Detailed Assumptions:

- o Clearly outline all major assumptions made in the methodology section, particularly those regarding biomass productivity and drainage classification.

Please see our earlier reply where we explain that we have done that in the uncertainty and limitations section.

Discuss their potential impact on the results in a limitations section.

We thank the reviewer for this suggestion. We have added an uncertainty and limitations section outlining the major assumptions, as previously mentioned. Additionally, we conducted sensitivity analyses, which are now referenced in the text in lines 391-394 and lines 407-416

- Uncertainty Analysis:

- o Provide a brief uncertainty analysis that highlights the major sources of potential error and their expected impact on the study's conclusions in the discussion section. Use simple sensitivity analyses to illustrate these points if possible.

Thank you for this nice idea! We implemented a sensitivity analysis for both the nutrient status of boreal forests and the drainage status of grasslands, as noted in previous replies. Additionally, as mentioned in response to another comment, we incorporated a spatial regional emissions comparison, demonstrating that our estimates align well with those regional maps. In response to your suggestion, we have now included the uncertainty inherent in the IPCC emission values (± 1

standard error) in our emission estimates. This provides a valuable additional range, illustrating where the true emissions for each land use and country are likely to fall.

- **Comparative Analysis:**

- o Compare the study's emission estimates with those from national inventory submissions and discuss any significant differences. Highlight areas where the study's methods provide improved accuracy.

General reasons for our considerably higher estimate of total GHG emissions from agriculture and forestry on drained peatlands have been given in lines 316-340 of the revised manuscript. In the rebuttal letter, we would like to provide you with a detailed summary of country-specific reasons, see below. This information is largely contained in the revised manuscript. Given the space limit, we cannot provide a detailed breakdown for each country regarding improvements in emission reporting accuracy in the manuscript and have therefore opted to keep some of the country-specific remarks more general.

Specific countries with considerable differences for agriculture (Cropland &

Grassland): Hungary does not report any drained area; we used a national soil map as input data¹⁰ and arrived at ~300 kha for drained peat and peaty soils. For the Netherlands arrived our study as larger drained area and NIS Netherlands (2023) does not report CH₄ from ditches. For Poland CH₄ (from ditches) and N₂O have not been reported; furthermore very low, outdated Tier 1 CO₂ EFs from IPCC (2006) have been used for Grassland. Another remarkable detail in the reporting for Poland is between 2019 and 2020 a massive shift occurred from Cropland to Grassland: Cropland: 532,13 kha (2019) to 160kha (2020) and Grassland 147,33 kha (2019) to 761,69 kha (2020). The implemented IPCC Tier 1 EFs are Cropland= 5 t C ha⁻¹a⁻¹ and Grassland only= 0,25 t C ha⁻¹a⁻¹. Lithuania is not reporting CH₄ from ditches and implements generally high thresholds for organic soils: "*soil is classified as organic if it has peat layer not thinner than 40 cm or 60 cm of poorly decomposed peat (mainly mossfibres) in bogs. In addition to this, histic horizon must contain not less than 70 - 75 percent of organic matter by volume.*" Our study reveals clearly higher area of agriculture on drained peatland. Lithuania moreover uses outdated Tier 1 CO₂ EF from IPCC (2006). Estonia is not reporting CH₄ from ditches, uses outdated Tier 1 CO₂ EF from IPCC (2006) for Grassland, and only reports half of the N₂O emissions from agricultural area. For France, our study reveals clearly higher area of agriculture on drained organic soils.

Specific countries with considerable differences for Forest Land. Generally, our study yielded for Baltic and Scandinavian countries larger areas for Forest Land on drained organic soils which may call for better differentiation of drained and undrained Forest Land in this regions, as we currently can do in our dataset. However, this potential overestimation does not produce any significant hotspot in the map. Poland does not report CH₄ (from ditches) and N₂O, and used very low, outdated Tier 1 CO₂ EF from IPCC (2006). Lithuania does not report CH₄ from ditches, uses outdated Tier 1 CO₂ EF from IPCC (2006) and generally implemented high thresholds for organic soils: "*A soil is classified as organic if it has peat layer not thinner than 40 cm or 60 cm of poorly decomposed peat (mainly mossfibres) in bogs. In addition to this, histic horizon must contain not less than 70 - 75 percent of organic matter by volume.*" Our study revealed a clearly higher area of drained organic soils under Forest

¹⁰ Pásztor, L., Laborczi, A., Bakacsi, Z., Szabó, J., & Illés, G. (2018). Compilation of a national soil-type map for Hungary by sequential classification methods. *Geoderma*, 311, 93-108.

Land. Estonia does not report CH₄ from ditches and uses outdated Tier 1 CO₂ EF from IPCC (2006). For Germany, we assume a calculation error for CO₂ emissions in the NIS, since areas, EF's and coverage of GHG sources seems to be comprehensive. Ireland is using a CO₂ EF that is clearly lower (1.68 t C ha⁻¹a⁻¹) than the Tier 1 CO₂ EF for Temperate Forest Land (2.6 t C ha⁻¹a⁻¹). Latvia uses a very low country-specific CO₂ EF (0.52 t C ha⁻¹a⁻¹). For UK, Annex Table A 3.4.26 gives CO₂ EFs of 0.95 to -0.34 t CO₂-C ha⁻¹ a⁻¹, whereas in the CRF the implemented EF is much lower (0,01 t CO₂-C ha⁻¹ a⁻¹) - which lowers the reported emissions dramatically compared to the Tier 1 CO₂ EFs for temperate Forest Land on drained organic soils.

- Policy Implications:

- o Expand the discussion on policy implications to include specific recommendations for policymakers based on the identified hotspots. Emphasize the importance of targeted mitigation measures in these areas.

We agree with the reviewer that we should expand our discussion on policy implications and include recommendations based on the hotspots we found. Therefore, we included a extra paragraph starting in line 439-452 of the revised manuscript.

Reviewer #2 (Remarks to the Author):

Comments on Giersbergen et al. Nature Communications

This paper attempts to identify drained wetland GHG hot spots for the EU. The new component is the overlaying of drained wetland data and agricultural land-use. This makes the areas involved different than previous analyses. However, the authors use the emission factors from Wilson et al., and these have huge uncertainties. The authors discuss the uncertainties in their land-use data set but accept the EFs without question? There authors make attributions that are not correct and use terms that are not relevant. With some work this could be a useful addition, but I not sure Nature Communications is the right place.

Comments

Ln 11 The only thing that makes these emission hot spots is the attribution of EFs to a specific land-use drained wetland class that appears in these locations. There is nothing special about the EFs - they are Wilson's update of the IPCC's update. These has been shown to have a seasonal bias to them by He and Roulet. Since the seasonality is largely latitude-dependent, what implication does this bias have to this work?

We thank the reviewer for pointing at this question. Updating the EFs was not in the focus of this manuscript. Our aim was to present the first ever peatland GHG emission map for the EU/EU+ and how to develop such a map, to illustrate the hotspots and to stress the importance of targeting specific mitigation measures in those regions. Updating the EFs would not change this message, as most the Cropland and Grasslands are located at the 50-55 latitudes (temperate region) where most often winter emission data is included in the data used for the EFs.

We do not find the paper by He and Roulet relevant as they look at peat extraction and not cropland and only at CO₂ and not the other GHGs, however it would have the same estimation error as not measuring during wintertime. Recent research suggests that the water table is the most significant factor regulating the GHG emissions^{11,12} and also that there is no need to lower the current EFs for the northern latitudes¹³

Based on the comments by reviewer 1 we decided to step back from the Wilson et al., 2016 EF as they are only updated with the newer GWP for not rewetted peatlands but already outdated as there are new GWP factors since AR6 2021. Therefore, we used the latest IPCC EFs, including those for ditches and also included uncertainties for all GHG's and land uses and within the GWP (see methods, table 1).

11 Evans et al. 2021. Overriding water table control on managed peatland greenhouse gas emissions. <https://doi.org/10.1038/s41586-021-03523-1>

12 Jauhainen et al. (2023) Greenhouse gas emissions from drained organic forest soils – synthesizing data for site-specific emission factors for boreal and cool temperate regions. <https://doi.org/10.5194/bg-20-4819-2023>

13 Honkanen, H. et al. (2023) Minor effects of no-till treatment on GHG emissions of boreal cultivated peat soil. *Biogeochemistry* 167, 499–522. <https://doi.org/10.1007/s10533-023-01097-w>

Ln 84 McCain et al. never suggested this. They did not mention peatlands in the 2003 article. Maybe the sentence is poorly written, but the attribution is wrong. McCain and the idea of HSHM is fine.

Thank you for pointing this out, this was indeed mentioned erroneously here. We have removed it and rewritten the sentence accordingly:

In line 86-87 of the revised manuscript: “Already more than 20 years ago, McClain et al. (2003) suggested that easy wins for the climate could be achieved by first focusing on hotspot areas.”

Ln 151 and throughout Tipping pt? Why is this a tipping point? Table S2 is productivity by country. What are the forcing variables for a tipping point?

We changed the wording to “threshold value” as it’s not really a tipping point. The forcing variables to assess the height of the threshold value was based on the nutrient poor/ nutrient rich fraction of the total peatland area delineated by Martin and Couwenberg 2021 (both also being co-authors to this manuscript).

Ln 184 Evaluating your outcome against the methodology used to develop it? Is it an independent evaluation? More details are needed here as this will allow the reader to determine if the results are a significant addition or not.

This is a very good point of the reviewer, many thanks. We used the method of Martin and Couwenberg 2021, i.e. not UNFCCC/NIS reporting to develop the methodology. However, they still used the same IPCC EF and scientific and grey literature. As it hind towards not independent evaluation we excluded the comparison in the new. Additionally, we included now the regional emission maps to compare and spatially validate our outcomes in the manuscript. In our reply to an earlier comment by reviewer 1 the outcomes are also discussed.

Since you are using fixed EFs from others, the only thing being evaluated here is the land-use wetland drainage maps, which are correct. Are there no independent maps from other analyses one could use for evaluation? I am thinking of the detailed work done by Connolly in Ireland.

Good point, because of this we added a section in the revised manuscript (line 264-277 of the revised manuscript) about a comparison of regional spatial maps in two federal states in Germany with our emissions map (unfortunately we could not get hold of the map by Connolly and others). Both regional maps^{14,15} are developed with different emission factors and therefore are a indicator to evaluate our estimate on regional scale. Based on this assessment the cumulative emissions per region fall within the 1 standard error of the IPCC emission factor estimate of our emission map. Most often the regional assessment is a bit lower than the mean IPCC EF but on average and spatially quite in-line. So, we are quite confident that our map can be used to asses the regional variability. See manuscript and supplement figures for more detail where the supplement figure also link towards the uncertainty in the IPCC EF.

14 Reichelt, F. (2021) Treibhausgas-Emissionen aus organischen Böden in Brandenburg. Greifswald Moor Centrum-Schriftenreihe 02/2021 (Selbstverlag, ISSN 2627-910X), 11 S

15 Ministerium für Klimaschutz, Landwirtschaft, ländliche Räume und Umwelt Mecklenburg-Vorpommern. (2025). Strategie zum Schutz und zur Nutzung der Moore in Mecklenburg-Vorpommern. Schwerin, Germany. Retrieved from <https://www.regierung-mv.de/serviceassistent/download?id=1675720>

Ln 194-202 I Don't see why percentage matters here, as the countries are not the same size? It is the absolute areas in each country that contribute emissions. Using percentages is misleading. Ireland and Scandinavia have a higher percentage of undisturbed peatlands, but they both have a significant area of drained peatlands that are emitting GHGs. Sweden is the second or third largest country in Europe (I recognize Europe and the EU are not the same, but you have Britain listed in this paragraph), depending on whether you consider the Ukraine part of Europe. Conversely, some countries on your list are tiny, so their contribution is small.

We seized the reviewer's suggestion and have adapted the text accordingly. We now focus on the larger peatland-rich countries.

Ln 258 This is an interesting map. The map results are primarily from drained peatlands in crop and grassland, as driven by the EFs assigned to those two classes. There is a lot of uncertainty in the EFs. Did you do any sensitivity analysis using the range of EF values in each category?

We found the idea of sensitivity analyses good – many thanks for proposing this. We looked at effects on nutrient status classification in boreal forested peatlands and drainage level classification in temperate grasslands. They indicate that potential changes would still support our conclusions on under-reporting in NIS. This has been outlined in the discussion in line 391-394 of the revised manuscript

Next to this we checked, also based on earlier comments our data with two regional peatland GHG emission maps from Germany and added the following in the discussion section which also assesses the sensitivity of our data in line 407-414 of the revised manuscript

Next to this we made a low and high hotspot and emission map to capture the full range of uncertainty. However, it did not change much with respect to the location of hotspots, only the matrix numbers changed a bit - therefore it is not included in the main part of the manuscript, but in the supplement as figure S4. See below the results where left is -1 std error middle is IPCC EF and right is +1 error deviation. We could have expected more difference as the standard deviation per land use is not equal.

Ln 279 Not a huge number when overall emissions are considered. Also, you compare this with other sources, and one can assume these are equally uncertain. Emissions reporting is a bit of a game.

Yes, indeed there are several sources of error in the inventory process. With respect to peatlands, it also seems that often the responsible authorities are not authorities with expertise on peatlands. Therefore, we believe that our table 2 can be helpful by pointing out specific cases of

under-reporting that should be further studied by the countries in question. As the resources of the teams of the EU and UNFCCC reviewing the GHG inventories is limited, this analysis can also ease their work in assessing the quality of the inventories and suggesting improvements in the inventory development.

Ln 364-368 In the previous section, you were referring to 2050. However, this argument is based on the GWPs of rewetted replacing drained using 100-year GWPs. 2050 is 25 years away so the 20-year GWP would be more honest. The picture would look very different if you use 20-year GWPs. The methane component becomes much more critical. I think you are beginning to mix apples and oranges here by discussing EU 2050 climate policy and a 100-year GWP. I don't think anyone can seriously argue that peatland restoration and rewetting are not good ideas in the long run, but you have hooked your argument to EU climate policy, so stick to the relevant time horizons for the policy. The 20-year GWPs can be figured out just as quickly as the 100-year.

We would like to thank the reviewer for bringing up this element of the debate within the climate mitigation community. The approach of the current paper uses as much as possible state-of-the-art knowledge and convention. To our knowledge the 100-year horizon to guide LULUCF policy is common ground to accommodate legacy and different rotation cycles across land-uses and changes between land-uses. We clarified this in the discussion section.

Moreover, we would like to exchange a number of considerations related to the GWP time horizon topic here. Firstly, we agree with the reviewer that there are several time horizons both warming due to GHG and policy action can be evaluated in. We acknowledge also that methane has a much faster atmospheric turnover than carbon dioxide while it on shorter time spans, can cause higher radiative forcing per mol tropospheric gas than carbon dioxide.

Secondly, applying a 20-year GWP to all methane sources across Europe may have transformative consequences for policy in the land-use sector. Methane emission from LULUCF and agricultural activities are estimated to range 20-30 Mt CH₄ annually with livestock emission being most prominent¹⁶. Including this emission in the current NIS and emission reporting using a 20-year GWP between 82 and 87 t CO₂eq per t CH₄ would add some 1,400 Mt CO₂eq annually to the balance sheet. Anthropogenic methane emission in the EU, largely livestock related, would become the single most emitting sector and more important than energy supply, domestic transportation and industry, respectively. Hence, the reduction targets of EU climate policy would largely turn on reducing livestock methane emission probably starting in peatlands given their high externalities and lower profitability.

Thirdly, linked to this argument we propose to take the livestock density of EU+ countries, methane emission maps and agricultural emission maps into account. The peatland emission hotspots largely overlap with high emission density agriculture (predominantly due to methane and N₂O, and to a lesser extent from the high peatland emissions). A plausible scenario would be that a reduction in livestock and drainage ditch emission would partly be compensated by post-rewetting methane emission.

Finally, we would like to question whether radiative forcing across time is more appropriate than a fixed GWP conversion for land-use policies. Policy must remain effective beyond 2050. CO₂ emitted today and until 2050 induces radiative forcing in 2051, but also in 2100, in 2250 and so

¹⁶ Sanois et al. (preprint): Global Methane Budget 2000-2020 <https://essd.copernicus.org/preprints/essd-2024-115/essd-2024-115.pdf>

forth¹⁷. Given the methodology developed in our research future work can adjust GWP conventions and time horizons depending on standards agreed on in the future. We have rewritten a part of the discussion, accordingly.

Ln 376 No uncertainty in the attribution and magnitude of the EFs?

We fully agree with the point that the IPCC EF appears to lack uncertainty in our first version, as significant uncertainties do exist within the EF, see updated table 1. As we revised and reprocessed the maps, we incorporated one standard error into the emissions calculation, which is now also shown in Table 2. Although EF errors vary for each EF and could potentially lead to different hotspot areas, the emission and hotspot maps remained largely unchanged. Therefore, we didn't included the -1 and +1 standard error estimates in the mapping of the manuscript.

¹⁷ See argumentation in Günther et al. 2020 and Köbsch et al. (preprint) Koebisch, F., Günther, A., Huth, V., Sachs, T., Glatzel, S., & Jurasinski, G. The climate efficacy of peatland rewetting under high CH₄ emissions. Preprint

Reviewer #3 (Remarks to the Author):

The major claim of the paper is providing a detailed maps of EU peatland use and its associated GHGs, and current NIR to UNFCCC underreported the emissions. The paper presents a literature synthesis for both mapping and emission estimation although the methods both for land use classification and emission factors (EF) can not been deemed as novel.

Whereas various emission estimates for EU have been made before (earlier ones e.g. Byrne et al 2004 EU peatlands: Current Carbon Stocks and Trace Gas Fluxes), this study updates those estimations. One improvement to further classify the land use categories was made through the allocation of nutrient and drainage levels by biomass productivity. This has to be made due to lack of data, but my concern is it differs from the IPCC EF classifications (by minerogenic vs ombrogenic and mean water table depth of 30 cm) increasing uncertainties simply due to two classifications are used for the area and EF. Would this influence the EFs and the GHG budgets? To which extend we should further classify the land use without introducing extra uncertainty?

We thank the reviewer for pointing out these important questions! Updating the national emission estimates was not the main goal of this study (cf. Martin & Couwenberg 2021) but we primarily aimed at increasing the accuracy and accountability in EU GHG reporting from peatlands by providing the first detailed peatland GHG emissions map and at highlight the areas where most of the emissions occur.

Regarding uncertainty and further classification into subclasses within a land use category we think that this goes at the cost of EF uncertainty as the amount of data backing up the EF per subclass diminishes due to low numbers of measurements done in “niches”. However, aspects such as land prices and opportunity costs are also important parameters when prioritising certain regions on the way to achieving the EU climate targets, which ultimately imply the rewetting of all drained peatlands. Rewetting shallow drained Grasslands or nutrient poor Forest Land has a low emission reduction potential but can be economically wise (such lands are already low productive areas). In contrast, deep drained Grassland and nutrient rich Forest Land are economic more valuable and also require bigger landscape-scale changes, but the GHG emission reduction potential is much greater.

In addition, the difference in water table depth (WTD) is one of the important drivers for CO₂ (see Figure below¹⁸), CH₄ (see Figure¹⁹ below) and N₂O emissions and therefore for the EF. So, the distinction between at least two groups that are most frequently observed can help with the policy/economical trade-off even though it might increase the uncertainty in EFs. At least the IPCC 1 standard error do not overlap between the subclasses which gives trust that it does not introduce much extra uncertainty.

Next to this uncertainty with EFs, there is also uncertainty in our research when assigning if a Forest Land is nutrient poor or rich (in boreal region) and a Grassland is shallow or deep drained (in the temperate region), respectively. Therefore, we made a small sensitivity analyses and added it to the uncertainties and limitation section in the discussion. The potential maximum effect if all boreal forest peatland were assigned as nutrient poor or nutrient rich is -5.5 up to +20.0 Mt compared to our estimates. Whereas if all temperate grasslands were assigned as

18 Aben, R. C. H., van de Craats, D., Boonman, J., Peeters, S. H., Vriend, B., Boonman, C., ... van den Berg, M. (2024). CO₂ emissions of drained coastal peatlands in the Netherlands and potential emission reduction by water infiltration systems. *Biogeosciences*, 21(18), 4099–4118. doi:10.5194/bg-21-4099-2024

19 Buzacott, A. J. V., Kruijt, B., Bataille, L., van Giersbergen, Q., Heuts, T. S., Fritz, C., ... van der Velde, Y. (2024). Drivers and annual totals of methane emissions from Dutch peatlands. *Global Change Biology*, 30(12), e17590. doi:10.1111/gcb.17590

shallow drained or deep drained this will change the total emissions from -34 up to +15Mt. Furthermore, if we would use one EF for temperate drained grassland having the average of deep and shallow drained then we would end up 9 Mt lower than we estimated now. However, the grassland classification has been verified with the national estimates in Martin & Couwenberg 2021 so we would not expect such a big change. At least relative to the total emission of agricultural activity these changes would still result in a significant underestimate in the NIS 2023.

REDACTED

REDACTED

The EFs were directly taken from Wilson et al. 2016. There are more recent updates of EF for subcategories, for instance, for forests on organic soils e.g. Jauhiainen et al 2024 revised the EF and should be used to update the respective EFs.

This is also a very good point of the reviewer, many thanks. The Wilson et al., 2016 paper is not really an update of the IPCC EF as they only updated the GWP for drained peatlands (see above). We have now exclusively used EFs from IPCC 2014 to ensure consistency in defining hotspots across Europe. We acknowledge that the IPCC estimates are based on only 8 to 13 Forest Land sites, compared to the 22 to 35 sites used by Jauhiainen et al. (2024). Jauhiainen et al. provide potentially valuable new EFs, but the publication is quite novel and countries have not yet applied them in their GHG inventories. Implementing these new EFs would make comparisons with existing National Inventory Submissions (NIS) rather unfair. Moreover, the study (Jauhiainen et al.) reported only minor differences between the new literature and existing Tier 1 EFs. Therefore, it is unlikely that the hotspot analysis would change significantly even if these new EFs were used. This last point can be substantiated with the added Supplement Figure 3 that shows the effect of using low and high IPCC EF estimates (± 1 standard error) and that the hotspots do not change much even though Forest Land has a higher variance around the mean IPCC EF.

I would further strongly suggest the authors to have a clearly defined boundary for their fluxes in this study i.e. to define what fluxes are included in their emission estimations.

The reviewer is right in that we often see papers where this is unclear. For the sake of clarity, we updated Table 1 where all emissions are delineated and added an extra column for ditches to highlight these are also taken into account. We also made a methodical figure in the supplement (Figure S1) to make it clear how we estimated the emissions. Next to this we added a sentence about how we estimated the ditch emissions using the IPCC guidelines.

Line 132-135: *“Peatland drainage requires ditches, so the emissions from these ditches must be included as well. Due to limited EU-wide data, default IPCC fractions of $5\% \pm 2.5$ for agricultural fields and $2.5\% \pm 1.25$ for Forest Land were applied (IPCC 2014). We did not include GHG emissions from (wild)fires.”*

Would emissions from aquatic fluxes included?

We seized the reviewer's suggestion and have made it more clear which fluxes are included. Please also refer to our answer of the comment above.

One aspect that is totally ignored fire emission. Drained peatlands are known to have fire probability and with climate change the risk is higher, which can cause events-based C fluxes. Fire emissions are included in the equations of Wilson et al 2016 for completion but no data were really incorporated in those EFs.

We neglected emissions from wildfires because the estimated values would be very difficult to compare to the reported values as they are not reported per soil type in the national GHG inventories. We explicitly now mention this to comply with the earlier comment of clearly defined boundaries.

References

Christensen TR, Friborg T (lead authors) with Byrne KA, Chojnicki B, Drösler M, Freibauer A, Frolking S, Lindroth A, Mailhammer J, Malmer N, Selin S, Turunen J, Valentini R, Zetterberg L, Vandewalle M. 2004. EU peatlands: Current carbon stocks and trace gas fluxes, Report 4/2004 to 'Concerted action: Synthesis of the European Greenhouse Gas Budget', Geosphere-Biosphere Centre, Univ. of Lund, Sweden

Jauhiainen, J., Heikkinen, J., Clarke, N., He, H., Dalsgaard, L., Minkinen, K., Ojanen, P., Vesterdal, L., Alm, J., Butlers, A., Callesen, I., Jordan, S., Lohila, A., Mander, Ü., Óskarsson, H., Sigurdsson, B. D., Søgaard, G., Soosaar, K., Kasimir, Å., Bjarnadottir, B., Lazdins, A., and Laiho, R.: Reviews and syntheses: Greenhouse gas emissions from drained organic forest soils – synthesizing data for site-specific emission factors for boreal and cool temperate regions, *Biogeosciences*, 20, 4819–4839, , 2023